# Time-uniform and Asymptotic Confidence Sequence of Quantile under Local Differential Privacy

**Leheng Cai**[1†], **Qirui Hu**[2,3†,∗] **Juntao Sun**[2†], **Shuyuan Wu**[2†]

[1] Department of Statistics and Data Science
Tsinghua University, Beijing, China

[2] School of Statistics and Data Science
Shanghai University of Finance and Economics, Shanghai, China

[3]Institute of Big Data Research, Shanghai University of Finance and Economics, Shanghai, China

cailh22@mails.tsinghua.edu.cn
huqirui@mail.shufe.edu.cn
sunjuntao1@stu.sufe.edu.cn
wushuyuan@mail.sufe.edu.cn

## Abstract

In this paper, we develop a novel algorithm for constructing time-uniform, asymptotic confidence sequences for quantiles under local differential privacy (LDP). The procedure combines dynamically chained parallel stochastic gradient descent (P-SGD) with a randomized response mechanism, thereby guaranteeing privacy protection while simultaneously estimating the target quantile and its variance. A strong Gaussian approximation for the proposed estimator yields asymptotically anytime-valid confidence sequences whose widths obey the law of the iterated logarithm (LIL). Moreover, the method is fully online, offering high computational efficiency and requiring only $\mathcal{O}(\kappa)$ memory, where $\kappa$ denotes the number of chains and is much smaller than the sample size. Rigorous mathematical proofs and extensive numerical experiments demonstrate the theoretical soundness and practical effectiveness of the algorithm.

## 1 Introduction

Mobile sensor traces (accelerometer, gyroscope, wireless-charger emissions) can be reverse-engineered to reveal routes, speech, and browsing habits [30, 57, 3, 37]. This shows privacy risks arise whenever fine-grained data are aggregated and mined; Differential Privacy (DP) mitigates this by adding calibrated noise so any individual's presence has negligible effect [22]. But DP assumes a trusted central curator, a model broken by the Netflix deanonymization and the March 2023 ChatGPT exposure [41, 34]. Local Differential Privacy (LDP) removes that single point of failure by randomizing data on-device and is already used in Google's RAPPOR, Apple's iOS telemetry, and Microsoft's Windows diagnostics. [20, 24, 17] As data ecosystems grow, LDP shifts analytics toward provable, user-centric privacy.

Quantile estimation and inference play a critical role in a variety of scientific and practical fields. In finance, quantiles such as value-at-risk and expected shortfall help manage portfolio risks under regulatory requirements and market volatility [10, 4]. Accurate estimation of extreme quantiles is especially important for capturing heavy-tailed financial risks [51]. In healthcare, quantile methods identify clinically significant thresholds, such as safe medication doses and treatment effectiveness

---

∗Corresponding Author

†The authors are listed in alphabetical order with equal contribution

39th Conference on Neural Information Processing Systems (NeurIPS 2025).

[55], and guide resource allocation in treatment prioritization [56]. Reliability engineering also frequently employs quantile estimation to establish conservative safety standards for machinery in harsh operating conditions [18, 29]. In addition, policy evaluation also benefits from quantile approaches to capture intervention effects across diverse population groups, highlighting impacts that mean-based analyses may miss [11, 33, 15]. Unlike traditional methods focused on averages, quantile-based methods are robust when dealing with skewed or heavy-tailed real-world data, thus providing deeper insight into complex data distributions [9]. More discussion can be found in [31].

A substantial body of literature addresses quantile estimation under either CDP or LDP. Early contributions in the CDP setting include [23, 36]. More recent work, such as [48], proposes a rate-optimal sample-quantile estimator that avoids histogram evaluation, and [26] extends this line of research to the simultaneous estimation of multiple quantiles. Quantile estimation under CDP remains an active topic, with applications ranging from bounded-support data [2] to large-scale query systems [5]. In online scenarios such as continual observation [21], algorithms can compress or recompute the added noise at each time step to improve efficiency [50]. Both cases require access to raw data and apply the privacy mechanism iteratively. In the LDP setting, the curator never observes raw data but only privacy-protected reports supplied by users. This constraint makes it considerably more challenging to design algorithms that achieve accurate quantile estimation while supporting rigorous statistical inference; for example, [38] proposed an SGD-based estimator, [39] studied inference for simultaneous quantiles, [7] investigated federated quantile inference, and [1] considered hierarchical mechanisms and noisy binary search.

Inference on quantiles under an LDP constraint is challenging because it requires estimating the asymptotic variance (or other normalizing constants) of the LDP quantile estimator. Classical central limit theorem results show that the efficiency of a quantile estimator hinges on the density value at the true quantile. For SGD-based methods, however, this density is difficult to recover using only the iterates or perturbed gradients. Moreover, estimating the Hessian matrix is non-trivial, owing to the non-smoothness of the quantile loss, even if one is willing to spend additional privacy budget. Pointwise confidence intervals can be built via self-normalization or random-scaling techniques, but asymptotic sequential inference requires an almost surely consistent variance estimator; see [52]. Recently, [58] developed a high-confidence inference framework using P-SGD with identical initial values across chains, thus obtaining an i.i.d. sequence. In their theoretical results, the number of chains is fixed and cannot ensure the consistency of the variance estimator. Inspired by this smart approach, we consider a dynamically chained P-SGD whose number of chains grows with the sample size to ensure variance consistency.

We highlight our contributions as follows:

(i) We develop a novel algorithm based on the dynamically chained P-SGD for constructing time-uniform, asymptotic confidence sequences for quantiles under LDP. The procedure operates fully online, offering high computational efficiency while requiring only $\mathcal{O}(\kappa)$ memory, where $\kappa$ is the number of chains and diverges to infinity at a rate much slower than $T$, e.g., at the order of $\log T$.

(ii) We derive an almost surely Gaussian approximation for the Polyak-Ruppert-type estimator of a quantile obtained by P-SGD. This result is non-trivial even in the non-private setting due to the non-smoothness of the loss function and its gradient. Notably, our strong Gaussian approximation is more general than those in [54] and [58], both of which address SGD with smooth loss functions. While the latter only establishes an $\mathcal{L}_2$ approximation for SGD within a fixed chain, which is not applicable to sequential inference. Our approximation rate is $\mathcal{O}_{a.s.}\big((T/\log\log T)^{-1/2}\big)$, faster than the LIL rate, yielding asymptotically anytime-valid confidence sequences for quantiles.

(iii) We propose an almost surely consistent estimator of the quantile variance that relies only on the iterates of P-SGD and incurs no additional privacy cost. Unlike [58], which uses a fixed number of chains, we allow the number of chains to grow with the sample size $T$ to ensure the consistency of variance estimation. As a by-product, the true density at the target quantile can also be consistently estimated under LDP. To the best of our knowledge, this is the first result on sequential inference for quantiles within the LDP framework.

The remainder of this paper is organized as follows. We first review the key concepts of DP and LDP. Next, we introduce our methodology, detailing the proposed algorithms together with the theoretical guarantees. Finally, we present experimental results that demonstrate the effectiveness of the approach. All theoretical proofs and additional simulation results are established in Appendix.

## 2 Methodologies

First, we introduce the mathematical definition of LDP and the asymptotic confidence sequence. Then, we will introduce the problem setting and algorithm details.

**Definition 1 (Differential Privacy , see [22])** *A randomized algorithm $\mathcal{A}$, taking a dataset consisting of individuals as its input, is $(\epsilon, \delta)$-differentially private if, for any pair of datasets $S$ and $S'$ that differ in the record of a single individual and any event $E$, satisfies the condition below:*

$$\mathbb{P}[\mathcal{A}(S) \in E] \leq e^\epsilon \mathbb{P}\left[\mathcal{A}\left(S'\right) \in E\right] + \delta.$$

*When $\delta = 0$, $\mathcal{A}$ is called $\epsilon$-differentially private ($\epsilon$-DP).*

**Definition 2 (Local Differential Privacy, see [32])** *An $(\epsilon, \delta)$-randomizer $R : X \to Y$ is an $(\epsilon, \delta)$-differentially private function taking a single data point as input.*

CDP regulates the distribution of released data rather than the curator's credibility. A trusted curator can centrally add noise, keeping algorithm design simple and accuracy loss modest [8].

LDP takes a stricter view by removing any trust assumption. The curator merely coordinates users, each holding a private value $X_i$. In each round it selects a user and specifies a randomized mechanism $R_i$. Users verify that the stated $(\epsilon, \delta)$ guarantee suits the study, apply $R_i$ to their data, and return the perturbed result. Interaction may be fully adaptive, sequential, or non-interactive; we adopt the tightest, non-adaptive model, fixing all user-randomizer pairs before data collection (Definitions 2.3 and 2.6 in [12]). Unlike CDP, where the curator adds noise, under LDP the curator must draw inference solely from user-randomized data.

From an inference perspective, the gap between a central-DP (CDP) estimator and its non-private analogue is typically $\mathcal{O}_p(n^{-1})$. Consequently, after $\sqrt{n}$ scaling, both estimators share the same asymptotic distribution, and one can estimate the associated variance from the (slightly perturbed) data by spending a modest additional privacy budget. By contrast, for locally private procedures, the error of an LDP estimator is usually $\mathcal{O}_p(n^{-1/2})$, which alters the limiting distribution and inflates the asymptotic variance. Moreover, in most practical settings, the variance cannot be consistently recovered from locally privatized data that were collected solely for point estimation.

**Definition 3 (Asymptotic confidence sequences, see [52])** *Let $\mathcal{T}$ be a totally ordered infinite set (denoting time) that has a minimum value $t_0 \in \mathcal{T}$. We say that the intervals $(\widehat{\theta}_t - L_t, \widehat{\theta}_t + U_t)_{t \in \mathcal{T}}$ centered at the estimators $(\widehat{\theta}_t)_{t \in \mathcal{T}}$ with non-zero bounds $L_t, U_t > 0, \forall t \in \mathcal{T}$, form a $(1 - \alpha)$-asymptotic confidence sequence (AsympCS) for a sequence of real parameters $(\theta_t)_{t \in \mathcal{T}}$ if there exists a (typically unknown) non-asymptotic $(1-\alpha)$-CS $(\widehat{\theta}_t - L_t^\star, \widehat{\theta}_t + U_t^\star)_{t \in \mathcal{T}}$ for $(\theta_t)_{t \in \mathcal{T}}$ —, i.e. satisfying*

$$\mathbb{P}\left(\forall t \in \mathcal{T}, \theta_t \in \left[\widehat{\theta}_t - L_t^\star, \widehat{\theta}_t + U_t^\star\right]\right) \geqslant 1 - \alpha,$$

*and $L_t, U_t$ become arbitrarily precise almost-sure approximations to $L_t^\star$ and $U_t^\star$ :*

$$L_t^\star / L_t \xrightarrow{a.s.} 1 \quad and \quad U_t^\star / U_t \xrightarrow{a.s.} 1.$$

Compared with classical asymptotic confidence intervals, AsympCS offer several advantages and have therefore attracted considerable research attention; see, for example, [40, 28, 27]. AsympCS quantifies uncertainty uniformly over all sample sizes, rather than at a single, pre-specified size. To guarantee valid coverage across this entire time horizon, the requisite consistency must hold almost surely, rather than merely in probability, as emphasized by [52].

We formulate the problem as follows. Let $\{\xi_t\}_{t=1}^T$ be independent observations drawn sequentially from a distribution $F$. Our goal is to construct an AsympCS for the $\tau$-quantile of $F$, denoted by $x^*$, i.e. $F(x^*) = \tau$, under a LDP framework.

To privatize $\{\xi_t\}_{t=1}^T$ we adopt the interactive, permutation-based binary-response mechanism of [38], which is optimal in certain regimes. Let $W_t$ and $V_t$ be i.i.d. Bernoulli variables, mutually independent and also independent of $\xi_t$, with

$$\mathbb{P}(W_t = 1) = r, \quad \mathbb{P}(W_t = 0) = 1 - r, \qquad \mathbb{P}(V_t = 1) = \mathbb{P}(V_t = 0) = 1/2.$$

For any $\zeta = (\xi, W, V)^\top$ and scalar $x$, define

$$G(x, \zeta) = \frac{1 + r - 2r\tau}{2} \Big[ \mathbf{1}\{\xi \leq x\} W + (1-W)(1-V) \Big] - \frac{1 - r + 2r\tau}{2} \Big[ \mathbf{1}\{\xi > x\} W + (1-W)V \Big].$$

Given a sequence $\{x_t\}_{t=1}^T$, this yields the privatized sequence $\{G(x_t, \zeta_t)\}_{t=1}^T$, which can be viewed as a permuted stochastic gradient. The parameter $r$ is the truthful-response rate, and [38] shows that the mechanism is $\epsilon$-LDP with $\epsilon = \log(1 + r) - \log(1 - r)$.

Using the privatized gradients, we run the SGD iteration

$$x_{t+1} = x_t - \eta_t \, G(x_t, \zeta_{t+1}), \qquad t = 0, \ldots, T-1.$$

Although this approach yields a consistent LDP estimator of the target quantile, estimating its asymptotic variance from $\{G(x_t, \zeta_t)\}_{i=1}^T$ alone is difficult. To address this, we employ parallel SGD (P-SGD): the data are split into $\kappa$ disjoint chains, all initialized identically,

$$x_{k,t+1} = x_{k,t} - \eta_t \, G(x_{k,t}, \zeta_{k,t+1}), \qquad t = 0, \ldots, T_k - 1, \ k = 1, \ldots, \kappa. \tag{1}$$

When each chain has the same length, the trajectories $\{x_{k,t}\}_{t=1}^{T_k}$ are i.i.d. across $k$, allowing the asymptotic variance to be estimated by the sample variance across chains. Ensuring consistency, however, requires $\kappa \to \infty$. Repartitioning the data would disrupt the SGD structure and consume additional privacy budget, so we adopt a dynamically chained P-SGD in which $\kappa$ grows with $T$.

To accommodate a time-varying number of chains, we let $\kappa = h(T)$, where $h : \mathbb{Z}_+ \to \mathbb{Z}_+$ is an increasing, piecewise-constant function. Set $K_0 := h(1)$. For each $k \in \mathbb{N}$ define $m_k := \big|\{T : h(T) = K_0 + k\}\big|$, where $|\cdot|$ denotes cardinality. We require

$$m_0 \geq K_0 \quad \text{and} \quad m_k \geq \frac{1}{K_0 + k - 1} \sum_{i=0}^{k-1} m_i, \qquad k \in \mathbb{Z}_+.$$

This condition ensures that no new chain will be added before the new chain is aligned in length with the previous ones. For example, $h(T) = \lfloor c \log_a T \rfloor + K_0$ with $a^{1/c} > \max\{K_0^{-1} + 2, \, K_0\}$ satisfies these conditions. Algorithm 1 provides the index of the chain to which each sample from 1 to $T$ is assigned. Figure 1 provides a visual illustration.

When the $T$-th sample arrives, let $T_k$ denote the number of observations held by the $k$-th chain. Our online quantile estimator is

$$\widehat{x}_T = \frac{1}{T} \sum_{k=1}^\kappa \sum_{t=1}^{T_k} x_{k,t} = \sum_{k=1}^\kappa \frac{T_k}{T} \Big( \frac{1}{T_k} \sum_{t=1}^{T_k} x_{k,t} \Big), \tag{2}$$

i.e., a weighted average of the chain-wise means. The asymptotic variance $\sigma^2$ of the approximating Gaussian variables $Z_i$'s in Theorem 1 is estimated by the weighted sample variance

$$\widehat{\sigma}_T^2 = \sum_{k=1}^\kappa \frac{T_k}{T} \Big[ \Big( T_k^{-1/2} \sum_{t=1}^{T_k} x_{k,t} \Big) - \sum_{l=1}^\kappa \frac{T_l}{T} \Big( T_l^{-1/2} \sum_{t=1}^{T_l} x_{l,t} \Big) \Big]^2. \tag{3}$$

Because both the quantile estimator (2) and the corresponding variance estimator (3) are computed directly from the P-SGD iterates in (1), they each satisfy $\epsilon$-LDP with $\epsilon = \log(1 + r) - \log(1 - r)$.

## 3  Theoretical results

To investigate the asymptotic properties, some mild assumptions are introduced.

(A1) The density $f(\cdot)$ is continuous and $f(x^*) > 0$.

(A2) For some constant $C_{f'} > 0$, $|f'(\cdot)|$ is uniformly bounded by $C_{f'}$.

(A3) For some constant $a \in (1/2, 1)$, the step size $\eta_t \asymp t^{-a}$.

(A4) As $T \to \infty$, $\kappa \to \infty$ and $\kappa \ll T^{1-1/(2a)}$.

---

**Algorithm 1** Data allocation for parallel runs

---

1: Input $T$ and function $h(\cdot)$.
2: Initialize array `nums` of length $\kappa_0 = h(0)$ with all zeros
3: Initialize array `result` of length $T$ with all zeros
4: **for** $i = 1$ to $T$ **do**
5:    **if** $h(i) > h(i-1)$ **then**    Append $0$ to the end of `nums`
6:    **end if**
7:    $k \leftarrow$ index of the first minimum in `nums`;  `result`$[i] \leftarrow k$;  `nums`$[k] \leftarrow$ `nums`$[k] + 1$
8: **end for**
9:    Output `result`

---

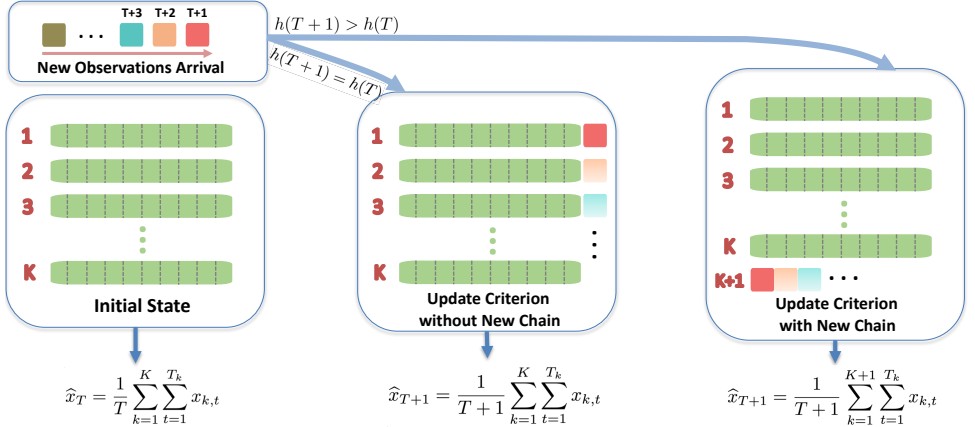

Figure 1: Overview of the Algorithm 1. (1) The left panel illustrates the initial state with $T$ observations partitioned into $K$ chains. (2) When new observations arrive, the algorithm determines whether to introduce a new chain. If not, new observations are sequentially added to existing chains, as shown in the middle panel. (3) If required, a new chain is created, as illustrated in the right panel, which continues receiving observations until it matches the length of existing chains. The update criterion ensures no additional chain is required before alignment.

Assumptions (A1) and (A2) are regular conditions for the distribution function. Assumption (A3) is standard in the literature; see [54]. Assumption (A4) restricts the rate at which the number of chains diverges with the sample size. The divergence rate can be arbitrarily slow, which offers great flexibility in practical implementation.

**Theorem 1** *Under Assumptions (A1)-(A4), for the quantile estimator (2) there exist i.i.d. normal r.v.'s $Z_i$'s with mean zero and variance $\sigma^2 = (1 - r^2(2\tau - 1)^2)/(4r^2 f^2(x^*))$, such that*

$$\left| \widehat{x}_T - \frac{1}{T} \sum_{i=1}^{T} Z_i \right| = \mathcal{O}_{a.s.}\left( \sqrt{\frac{\log\log T}{T}} \right).$$

Theorem 1 establishes a strong Gaussian approximation for $\widehat{x}_T - x^*$, providing an almost surely result rather than one in probability. Interestingly, for each fixed $k$, although $x_{t,k}$ are dependent across $t$, the deviation of the final weighted sum estimator $\widehat{x}_T$ from the true value $x^*$ can be approximated by the average of $T$ i.i.d. Gaussian random variables. Besides, the rate is significantly faster than the law of iterated logarithm bound, which is crucial for constructing asymptotic confidence sequences. Notably, [58] derived a Gaussian approximation result for a single chain, but their approximation error is measured in terms of mean squared error rather than an almost surely bound, which is not applicable to sequential inference. On the other hand, although [54] provides an almost surely Gaussian approximation, both [54] and [58] consider smooth loss and assume average Lipschitzness of the gradient, which does not hold for the quantile loss. In fact, the gradient of quantile loss only enjoys average 1/2-Hölder smoothness, since $\left[\mathbb{E}\{\mathbf{1}(\xi \leq x) - \mathbf{1}(\xi \leq y)\}^2\right]^{1/2} \lesssim |x - y|^{1/2}$ for any random variable $\xi$ with uniformly bounded density functions, which poses challenges for theoretically analyzing the approximation error.

The next theorem shows the almost surely consistency of $\widehat{\sigma}_T^2$.

**Theorem 2** *Under Assumptions (A1)-(A4), for the variance estimator (3) as $T \to \infty$,*

$$\left| \widehat{\sigma}_T^2 - \frac{1 - r^2(2\tau - 1)^2}{4r^2 f^2(x^*)} \right| = \mathcal{O}_{a.s.}(1).$$

A byproduct of Theorem 2 is that it enables the estimation of the density at the true quantile $x^*$ under the framework of differential privacy, i.e., $\sqrt{\{1 - r^2(2\tau - 1)^2/(4r^2\widehat{\sigma}_T^2)\}}$.

Let $[\mu_T - \gamma_{T,m}, \mu_T + \gamma_{T,m}]_{T \geq m}$ be any confidence sequence started from time $m \geq 1$ for the unknown mean of a Gaussian distribution with unit variance.

**Theorem 3** *Under Assumptions (A1)-(A4), there exists some nonasymptotic $(1 - \alpha)$-confidence sequence $[\widehat{x}_T - \sigma\gamma_{T,m}^\star, \widehat{x}_T + \sigma\gamma_{T,m}^\star]$, i.e., $\mathbb{P}\left(\forall T \geq m, x^* \in [\widehat{x}_T - \sigma\gamma_{T,m}^\star, \widehat{x}_T + \sigma\gamma_{T,m}^\star]\right) \geq 1 - \alpha$, such that $(\sigma\gamma_{T,m}^\star)/(\widehat{\sigma}_T\gamma_{T,m}) = \mathcal{O}_{a.s.}(1)$ as $T \to \infty$.*

With the help of the strong consistency established in Theorem 2, Theorem 3 provides a general framework for constructing AsympCSs for quantiles under the LDP setting, requiring only a confidence sequence for Gaussian random variables with unit variance. Existing confidence sequences for Gaussian variables in the literature include different types of boundaries, for example, the stitched boundary developed by [28] with a concentration rate of $\mathcal{O}\left(\sqrt{\log\log T/T}\right)$:

$$\gamma_{T,m} = 1.7\sqrt{\frac{\log\log(\max\{2T/m, e\}) + 0.72\log(10.4/\alpha)}{T}}, \tag{4}$$

or Robbins' mixture boundary ([44] and [45]), which achieves a concentration rate of $\mathcal{O}(\sqrt{\log T/T})$:

$$\gamma_{T,m} = \sqrt{\frac{\{g^{-1}(\alpha)\}^2 + \log(T/m)}{T}}, \quad \text{where } g(a) = 2\{1 - \Phi(a) + a\phi(a)\}. \tag{5}$$

Here, $\Phi(\cdot)$ and $\phi(\cdot)$ are the CDF and PDF of a standard Gaussian random variable, respectively. For $m = 1$, the Gaussian mixture bound can be generalized to the following, see [52],

$$\gamma_{T,1} = \sqrt{\frac{2(T\rho^2 + 1)}{T^2\rho^2}\log\left(\frac{\sqrt{T\rho^2 + 1}}{\alpha}\right)}, \quad \forall \rho > 0. \tag{6}$$

By tuning the hyperparameter $\rho$ in (6), one can minimize the width of the confidence interval at a specific time point given a significance level $\alpha$.

We note that the Robbins' boundary is not inferior due to its slower asymptotic convergence rate. On the contrary, it is often preferable in practice because it tends to be tighter in early stages with finite samples, as also discussed in [52].

There may be some confusion regarding the burn-in strategy used in SGD-based methods versus the construction of AsympCSs starting from index $m$. The burn-in strategy discards a predetermined number of initial iterates to mitigate the effect of unstable early updates on the final averaged estimator, thereby reducing the effective sample size. In contrast, the coverage probability calculation starting from $m$ retains all iterates from 1 to $m$ and uses them to construct the AsympCSs based on equations (4)-(5). If a burn-in of $b$ iterations is applied and coverage probabilities are reported starting from index $m$, then the AsympCSs should begin at iteration $b + m + 1$.

Combined with estimators (2), (3) and Theorems 3 ,we summarize the construction of the LDP $(1 - \alpha)$-AsympCS in Algorithm 2. It is worth noting that the entire procedure can be computed sequentially, storing only the most recent updates from each chain, thereby requiring approximately $\mathcal{O}(\kappa)$ memory, where $\kappa \ll T$. As a straightforward derivation, following Theorems 1 and 2, the LDP point-wise confidence interval of quantile is concluded as follows.

**Corollary 1** *Under Assumptions (A1)-(A4), the asymptotically correct $(1 - \alpha)$ point-wise confidence interval of quantile $x^*$ is $[\widehat{x}_T - \widehat{\sigma}_T z_{1-\alpha/2}/\sqrt{T}, \widehat{x}_T + \widehat{\sigma}_T z_{1-\alpha/2}/\sqrt{T}]$, i.e., $\mathbb{P}\left(x^* \in [\widehat{x}_T - \widehat{\sigma}z_{1-\alpha/2}/\sqrt{T}, \widehat{x}_T + \widehat{\sigma}z_{1-\alpha/2}/\sqrt{T}]\right) \geq 1 - \alpha$, as $T \to \infty$, where $z_{1-\alpha/2}$ is the $(1 - \alpha/2)$-quantile of standard normal random variables.*

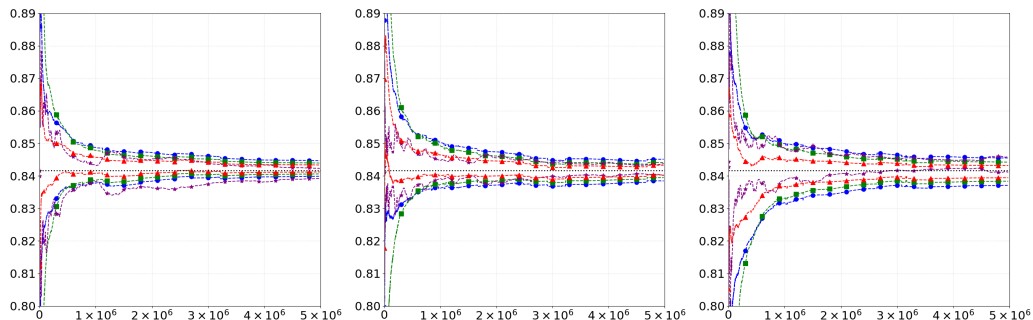

Figure 2: Plots of trajectories when confidential data come from standard normal distribution $\mathcal{N}(0,1)$ for pointwise confidence interval from Corollary 1 (in red with upward-pointing triangles), pointwise confidence interval from [38] (in purple with asterisks), proposed AsympCS based on (4) (in blue with circles) and (6) (in green with squares) with $\tau = 0.8$, $r = 1, 0.9, 0.75$ (left, middle and right panel).

It is well known that the tail of a self-normalized distribution is typically heavier than that of the normal distribution, as noted in [47]. As a result, the pointwise confidence interval constructed based on Corollary 1 is more efficient than those proposed in [38]. We further provide a visualization comparing our constructed AsympCSs, the pointwise confidence intervals, and the intervals from [38] in Figures 2 and A.7. One can observe that although the asymptotic widths of both pointwise confidence intervals are similar, our proposed intervals tend to be slightly narrower. Additionally, the AsympCS constructed using equation (6) is numerically tighter than the one based on equation (4), although the latter enjoys a faster asymptotic convergence rate.

---

**Algorithm 2** Algorithm to construct LDP AsympCS of quantile

---

1: Input data: $\{\xi_t\}_{t=1}^T$, truthful response rate $r \in [0,1]$, significance level $\alpha$, initial sample size $m$ for sequential inference, initial number of chains $\kappa$, learning rate $\{\eta_t\}_{t=1}^T$, initial index $n_k = 0$ and initial values across all chains $\widetilde{x}_k = x_0$.
2: **For** $t = 1, \ldots T$,
3:     Computed the current update chain $l_t = \texttt{result}[t]$ from Algorithm 1.
4:     **If** $l_t > \kappa$     Set $\kappa = \kappa + 1$, $n_\kappa = 0$, $x_{\kappa, n_\kappa} = 0$.
5:     **EndIf**
6:     Require perturbed gradient $G(x_{l_t, n_{l_t}}, \zeta_t)$.
7:     Update in $l_t$- chain: $x_{l_t, n_{l_t}+1} = x_{l_t, n_{l_t}} - \eta_{n_{l_t}} G(x_{l_t, n_{l_t}}, \zeta_t)$,
8:     $\widetilde{x}_{l_t} = \left\{ n_{l_t} \widetilde{x}_{l_t} + x_{l_t, n_{l_t}+1} \right\} / (n_{l_t} + 1)$,  $n_{l_t} = n_{l_t} + 1$,
9:     Update of quantile estimator and corresponding variance estimator:

$$\widehat{x}_t = \sum_{k=1}^{\kappa} \frac{n_k}{t} \widetilde{x}_k, \quad \widehat{\sigma}_t^2 = \sum_{k=1}^{\kappa} \frac{n_k}{t} \left[ \left( n_k^{1/2} \widetilde{x}_k \right) - \sum_{l=1}^{\kappa} \frac{n_l}{t} \left( n_l^{1/2} \widetilde{x}_l \right) \right]^2.$$

10: **End For**
11: Output the $(1 - \alpha)$-AsympCS $[\widehat{x}_t - \widehat{\sigma}_t \gamma_{t,m}, \widehat{x}_t + \widehat{\sigma}_t \gamma_{t,m}]$ for $t = m, \ldots, T$, where $\gamma_{t,m}$ can be computed by (4), (5) or (6)

---

# 4 Experiments

## 4.1 General setting

In this section, we evaluate the finite-sample performance of the proposed method. The confidential data are generated from two distributions: standard Normal $\mathcal{N}(0,1)$ and standard Cauchy $\mathcal{C}(0,1)$. Target quantiles are set to $\tau = 0.8, 0.5, 0.3$. The truthful response rates are chosen as $r = 1, 0.9, 0.75, 0.5, 0.25$, corresponding to privacy budgets $\varepsilon = \log(1 + r) - \log(1 - r)$ of $+\infty, 2.94, 1.95, 1.10, 0.51$, respectively. The algorithm uses random initialization with standard

Normal $\mathcal{N}(0,1)$ of all chains and step sizes set to $\eta_{\kappa,t} = 1/t^a$ with $a = 0.6$ for all chains as well, satisfying Assumption (A3). Following [35], we incorporate a burn-in strategy into the algorithm to reduce the impact of initial parameter bias and enhance the stability of statistical inference, with the number of burn-in samples being about $(0.25/r^2)\%$ of the total sample size. Each experiment is replicated 2000 times using 110 Intel® Xeon® Platinum 8352V CPU @ 2.10GHz CPUs with $360GB$ memory and $1200GB$ storage.

### 4.2 Results

Our first analysis focuses on the time-uniform convergence performance. We consider the number of chains as a function of time via $h(t) = \lfloor 8 \log_{10}(T/5) \rfloor$ for $t < T/5$, and $h(t) = \lfloor 8 \log_{10}(t) \rfloor$ for $t \geq T/5$, where $T = 5,000,000$ denotes the total sample size. Time-uniform $95\%$ AsympCSs are constructed using the stitched boundary in (4) and the Gaussian mixture boundary in (6) with $\rho = 0.001$. We report the time-uniform type I error rates and the average lengths of the resulting CSs. As a benchmark, we include the order-statistics-based non-private method proposed in [27].

Results for the standard normal distribution based on equations (4) and (6) are presented in Figures 3 and 4, while additional results for the standard Cauchy distribution are provided in Appendix A. These numerical results are consistent with Theorem 3. Figure 3 shows that all methods maintain the nominal type I error rate (5%) across various values of the parameters $r$ and $\tau$ for both AsymCSs based on (4) and (6). Figure 4 indicates that the average length of the constructed AsympCSs decreases as the privacy budget increases. Moreover, From Figure 3, one observes that the AsymCSs based on (4) will be more conservative than (6) in most stages under finite sample sizes, which is also reflected on Figure 4. Therefore, the AsymCSs based on (6) enjoys the better finite sample performance in our setting, even its theoretically asymptotic rate is $\mathcal{O}(\sqrt{\log T/T})$, which is slower than $\mathcal{O}(\sqrt{\log \log T/T})$.

Notably, when $r = 1$, the non-DP AsympCSs based on P-SGD are tighter than the nonasymptotic CSs from [27], while still maintaining valid type I error control. These findings suggest that our proposed confidence sequences can provide improved efficiency for quantile inference, even in the absence of privacy constraints. A similar phenomenon is observed under the Cauchy distribution setting, as illustrated in Figures A.3 and A.4.

Next, we investigate finite-sample variance estimation $\widehat{\sigma}_T^2$. To illustrate consistency with respect to $T$, we set $\kappa = 20, 40, 80, 100$. Relative absolute errors (RAEs), defined as $|\widehat{\sigma}_T^2 - \sigma^2|/\sigma^2$, are summarized via boxplots in Figures A.5 and A.6 in Appendix A . The results demonstrate that RAEs consistently decrease as $T$ increases, aligning with Theorem 2. Furthermore, for a fixed $T$, smaller values of $r$ yield lower RAEs.

Finally, to further strengthen our simulation study, we conducted additional experiments, including: (1) sensitivity analysis of tuning parameters, (2) finite-sample performance under a mixture of Beta distributions, and (3) a comparison between our proposed method and [38] under specific settings. Across these settings, the results consistently demonstrate the robustness and effectiveness of our approach; see details in Appendix A.

## 5    Real data application

In this section, we empirically evaluate the effectiveness of our proposed method on the following two representative real datasets widely used in privacy research:

**Law school dataset [53].** This dataset consists of 20,649 examples aiming to predict students' undergraduate GPA based on their personal information and academic abilities. Given that GPA reflects individual educational outcomes and is protected under strict data-use agreements [53], we treat it as sensitive educational information requiring privacy protection.

**Government salary dataset [42].** This dataset originates from the 2018 American Community Survey conducted by the U.S. Census Bureau. It includes over 200,000 observations, with annual salary (USD) as the response variable. Annual salary represents typical personal financial information [26]; therefore, we treat it as sensitive data warranting privacy protection.

To facilitate analysis, we applied a logarithmic transformation for two datasets and then back-transformed the confidence sequence bounds after prediction. We apply our proposed method:

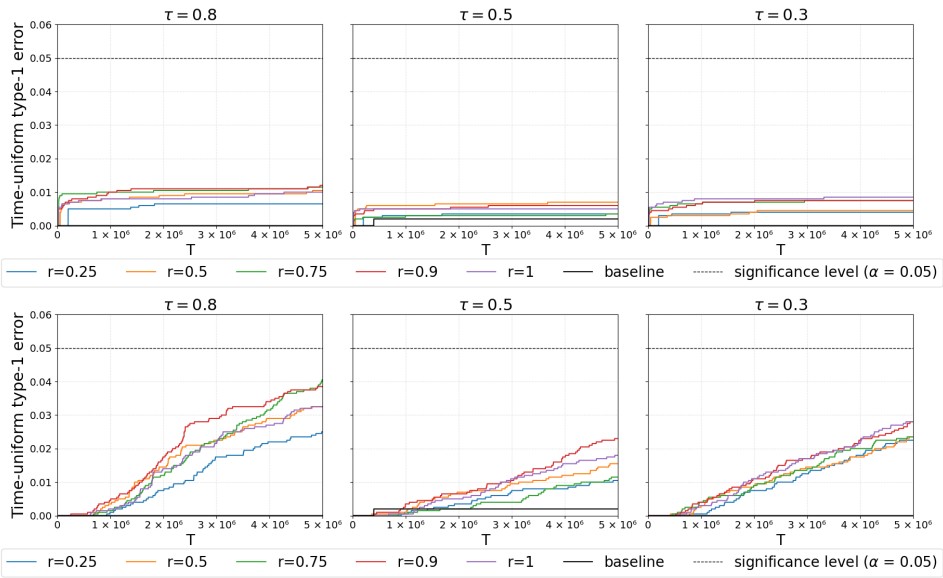

Figure 3: Time-uniform type I error for AsypmCS constructed by (4) (on top panel) and (6) (on bottom panel) and non-DP non asymptotic CS in [27], when confidential data come from standard Normal $\mathcal{N}(0,1)$ with $r = 0.25, 0.5, 0.75, 0.9, 1$ and $\tau = 0.3, 0.5, 0.8$.

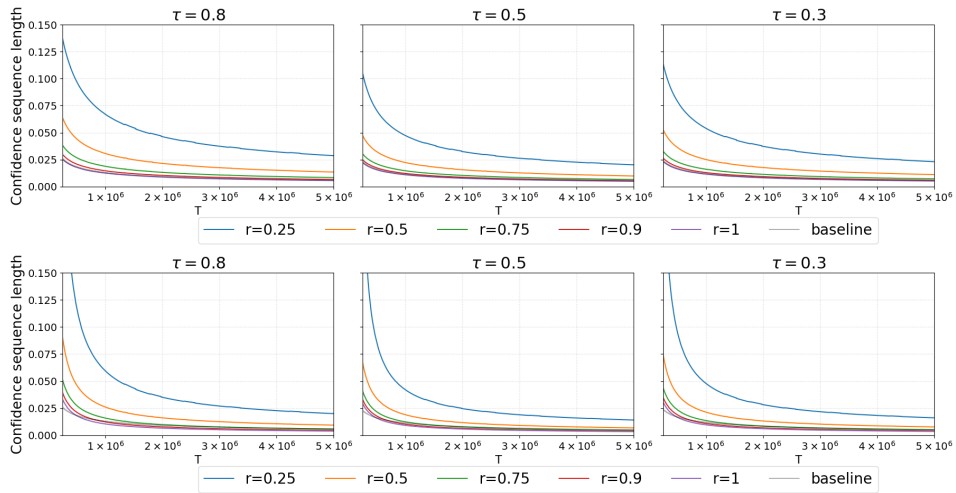

Figure 4: Average length for AsypmCS constructed by (4) (on top panel) and (6) (on bottom panel) and non-DP non asymptotic CS in [27], when confidential data come from standard Normal $\mathcal{N}(0,1)$ with $r = 0.25, 0.5, 0.75, 0.9, 1$ and $\tau = 0.3, 0.5, 0.8$.

AsympCS based on equation (4) and equation (6) to conduct privacy-preserving inference on the median ($\tau = 0.5$), targeting GPA in the first dataset and annual salary in the second. Specifically, we construct time-uniform CS for the respective quantiles under truthful response rates $r = 0.75$ and $0.9$, and set the hyperparameter $\rho$ in equation (6) to $\rho = 0.01$, while keeping all other tuning parameters consistent with those used in Section 4. The upper and lower bounds of the confidence sequences are presented in Figure 5. From the results in Figure 5, we observe that the CSs produced by our two methods under different response rates $r$ covering similar central values. In addition, in both datasets, the length of the constructed CS decreases as $r$ increases. As $t$ grows, the sequence based on (4) becomes more conservative than that based on (6). These observations align with our simulation findings and further demonstrate the methods' adaptability.

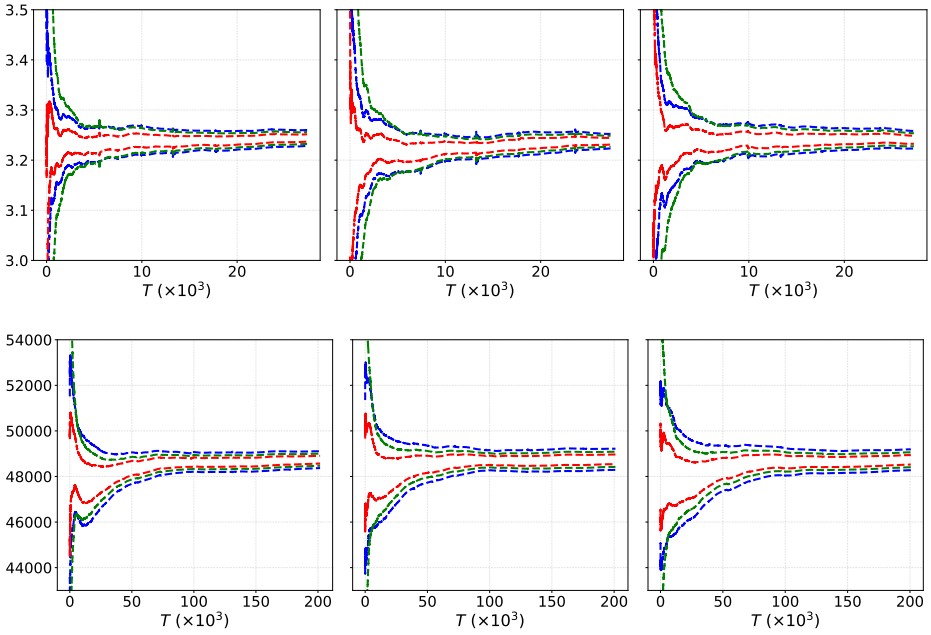

Figure 5: Confidence sequence boundaries for GPA in the Law dataset (on top panel) and annual salary in the government salary dataset (on bottom panel). The pointwise confidence interval from Corollary 1 (red), and the proposed AsympCS based on equation (4) (blue) and equation (6) (green), with target quantile $\tau = 0.5$ and truthful response rates $r = 0.9, 0.8$, and $0.75$ (left, middle, and right panels, respectively).

## 6   Concluding remark

In this paper, we introduce an online, $\mathcal{O}(\kappa)$-memory algorithm that provides time-uniform, asymptotic confidence sequences for quantiles under LDP. We establish an almost-sure Gaussian approximation for the Polyak-Ruppert quantile estimator obtained via parallel SGD, which is non-trivial even in the non-DP case at rate $\mathcal{O}_{a.s.}\big((T/\log\log T)^{-1/2}\big)$, thereby sharpening the $\mathcal{L}_2$ and smooth-loss based results of [58] and [54]. In addition, we devise an almost-surely consistent estimator of the quantile variance (and density) using only the SGD iterates, thus providing the first sequential quantile-inference procedure in the LDP setting.

Nonetheless, our methodology has some limitations. First, the SGD-based procedure depends on tuning parameters, such as the learning rate and the initial values, whose optimal calibration can be delicate. Second, the rate of variance consistency hinges on the number of parallel chains, $\kappa$, and the dynamically increasing-chain scheme requires relatively sharp assumptions on the relationship between $\kappa$ and $T$. A fixed-$\kappa$ variant could leverage a $t$-distribution with $(\kappa-1)$ degrees of freedom to form confidence sequences, but deriving non-asymptotic $t$-based bounds is far from straightforward. Finally, although non-asymptotic error bounds for SGD estimators have been extensively studied (e.g., 9), extending these results to obtain fully non-asymptotic confidence sequences for SGD iterates under LDP remains an attractive yet challenging avenue for future research.

## 7   Acknowledgments

The authors sincerely thank the anonymous reviewers, AC, and PCs for their valuable suggestions that have greatly improved the quality of our work. This work was supported by the Shanghai Engineering Research Center of Finance Intelligence (Grant No,19DZ2254600). Leheng Cai would thank to the funding supported by China Association for Science and Technology and National Natural Science Foundation of China (No. 12171269). Shuyuan Wu's research is partially supported by National Natural Science Foundation of China (No. 12401392) and China Postdoctoral Science Foundation (No. 2024M751929, No. 2024T170540).

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

# A Additional simulation results

## A.1 Other results in Section 4

In this section, we provide additional figures not shown in the main text (see Figures A.1–A.7). Notably, in the left panel of Figure A.1, the time-uniform type I error slightly exceeds the nominal level of 0.05 (approximately 0.08), with most errors occurring during the start of the algorithm. This primarily occurs because the initial values ($\mathcal{N}(0,1)$) are relatively far from the true values, resulting in poor estimation at the initial moments. Additionally, although (4) is generally more conservative than (6) during the early stage, calculations indicate that the boundary given by (4) is narrower at the very beginning of the algorithm, causing slightly poorer coverage during these initial moments. Increasing the burn-in period could further reduce this error rate.

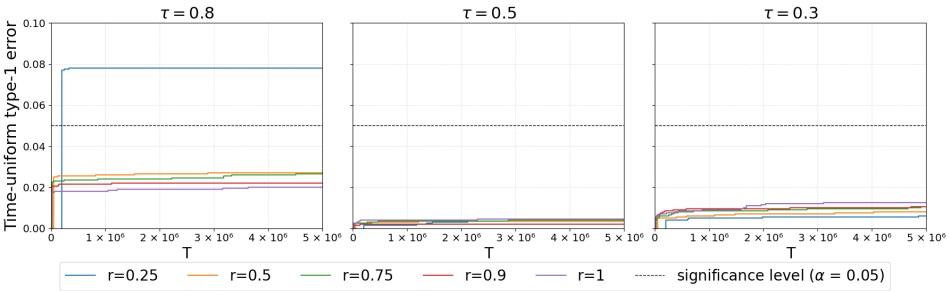

Figure A.1: Time-uniform type I error for AsypmCS constructed by (4) and non-DP non asymptotic CS in [27], when confidential data come from standard Cauchy $\mathcal{C}(0,1)$ with $r = 0.25, 0.5, 0.75, 1$ and $\tau = 0.3, 0.5, 0.8$.

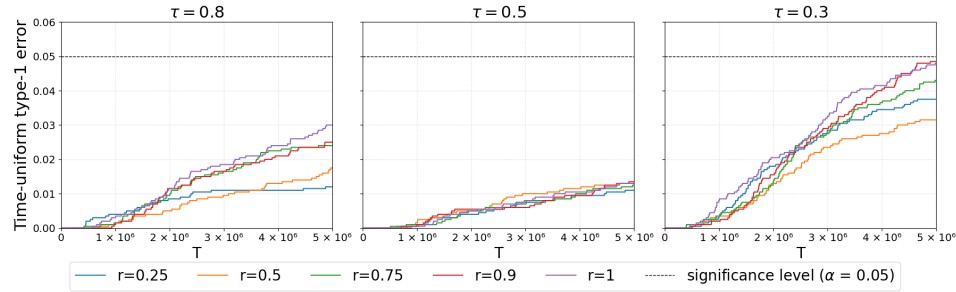

Figure A.2: Time-uniform type I error for AsypmCS constructed by (6) and non-DP non asymptotic CS in [27], when confidential data come from standard Cauchy $\mathcal{C}(0,1)$ with $r = 0.25, 0.5, 0.75, 0.9, 1$ and $\tau = 0.3, 0.5, 0.8$.

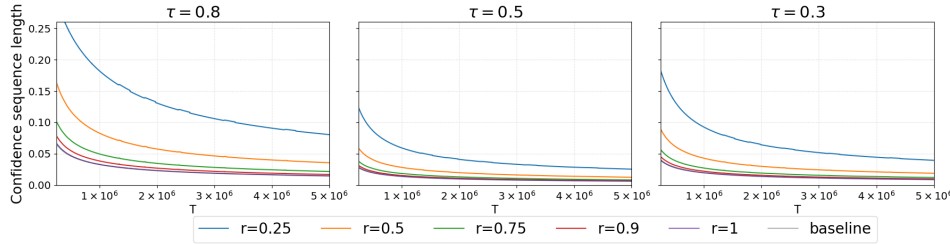

Figure A.3: Average length for AsypmCS constructed by (4) and non-DP non asymptotic CS in [27], when confidential data come from standard Cauchy $\mathcal{C}(0,1)$ with $r = 0.25, 0.5, 0.75, 0.9, 1$ and $\tau = 0.3, 0.5, 0.8$.

Next, we evaluate the finite-sample performance under a mixture of Beta distributions and make some discussions about our Assumptions. To be specific, for our Assumption (A1), according to

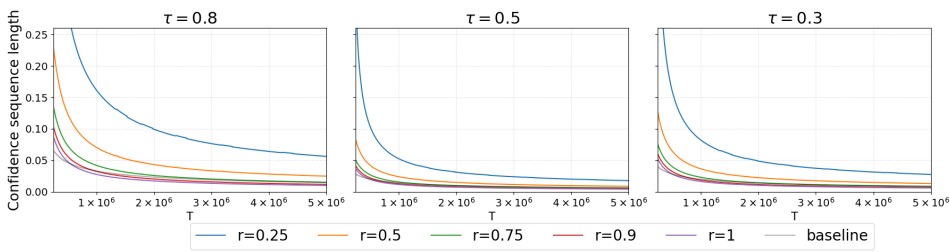

Figure A.4: Average length for AsypmCS constructed by (6) and non-DP non asymptotic CS in [27], when confidential data come from standard Cauchy $\mathcal{C}(0,1)$ with $r = 0.25, 0.5, 0.75, 0.9, 1$ and $\tau = 0.3, 0.5, 0.8$.

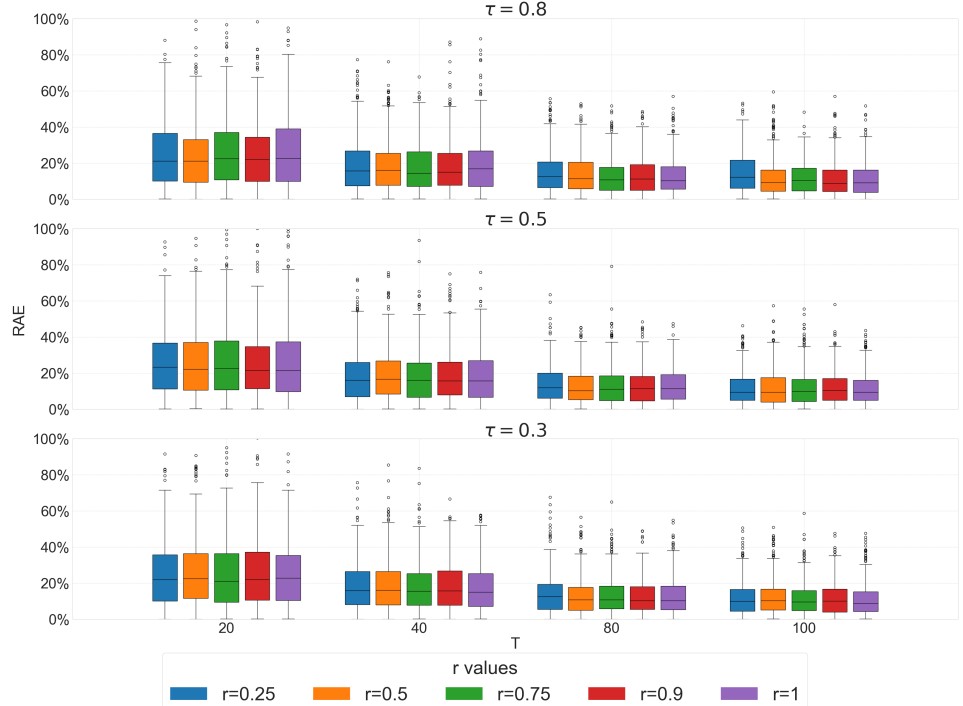

Figure A.5: Relative error of the variance estimator (3) when confidential data come from standard Normal $\mathcal{N}(0,1)$ with $r = 0.25, 0.5, 0.75, 0.9, 1$ and $\tau = 0.3, 0.5, 0.8$.

Corollary 2.3.3.A in [46], the asymptotic normality of the sample quantile relies on the assumption that the distribution function $F(\cdot)$ is differentiable at the true quantile value $x^*$, with a strictly positive derivative. While it's not strictly necessary that a density function $f(\cdot)$ equals the derivative of the distribution function, $f(\cdot) = F'(\cdot)$, this relationship holds if the density $f(\cdot)$ is continuous at $x^*$, in which case $F'(x^*) = f(x^*) > 0$. In addition, Assumption (A2) is a technical requirement crucial for controlling a negligible term in the Gaussian approximation, where a second-order Taylor expansion is applied (refer to equation 7). This is a mild assumption that holds for many common distributions, including heavy-tailed ones. For example, the derivative of the density function for a standard Cauchy random variable is:

$$f'(x) = \frac{d}{dx} f(x) = \frac{d}{dx} \left( \frac{1}{\pi(1+x^2)} \right) = -\frac{2x}{\pi(1+x^2)^2},$$

This derivative is both continuous and bounded, thereby satisfying Assumption (A2).

While Assumptions (A1) and (A2) hold for a wide range of distributions, our theoretical results require the underlying density to have continuous and bounded derivatives. This condition is not met by all irregular or "spiky" distributions. To investigate our method's practical performance under

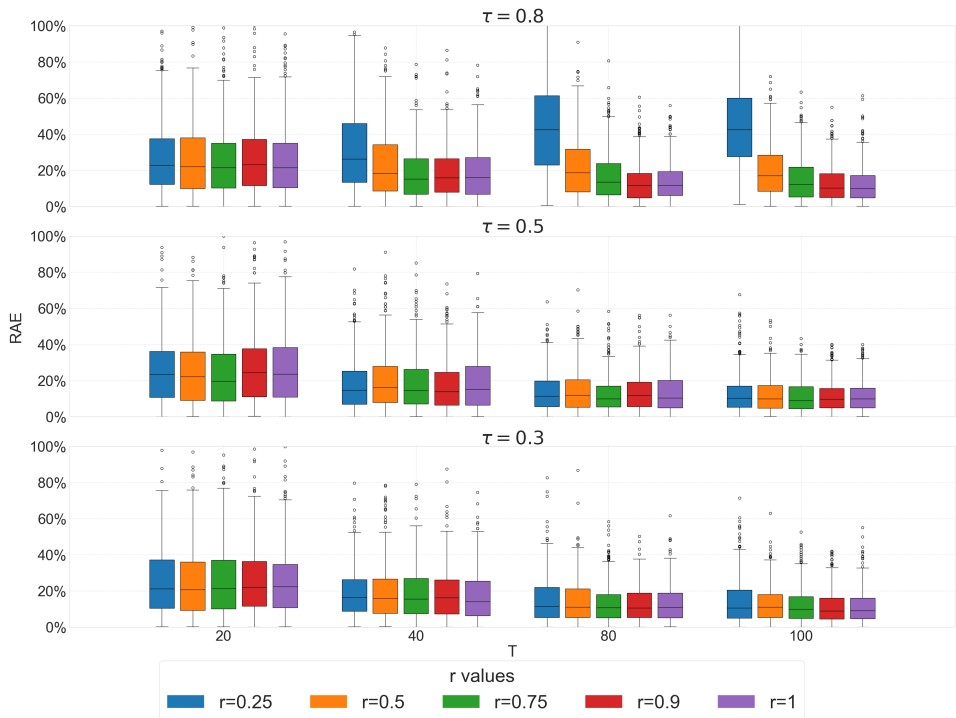

Figure A.6: Relative error of the variance estimator (3) when confidential data come from standard Cauchy $\mathcal{C}(0,1)$ with $r = 0.25, 0.5, 0.75, 0.9, 1$ and $\tau = 0.3, 0.5, 0.8$.

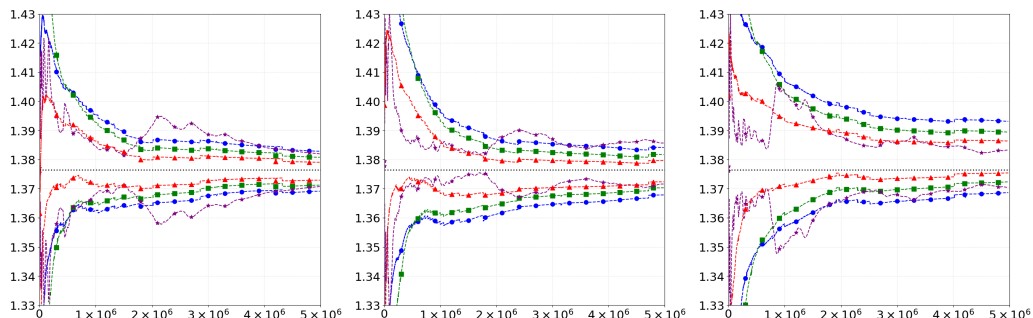

Figure A.7: Plots of trajectories when confidential data come from standard Cauchy distribution $\mathcal{C}(0,1)$, for pointwise confidence interval from Corollary 1 (in red with upward-pointing triangles), pointwise confidence interval from [38] (in purple with asterisks), proposed AsympCS based on (4) (in blue with circles) and (6) (in green with squares) with $\tau = 0.8$, $r = 1, 0.9, 0.75$ (left, middle and right panel).

these challenging conditions, we conducted further experiments. To be specific, we test our method using a mixture of Beta distributions with the following density:

$$f(x) = \{\beta_{10,100}(x) + \beta_{100,100}(x) + \beta_{100,10}(x)\} / 3.$$

where $\beta_{\alpha,\beta}(x)$ is the density of a Beta distribution with parameters $\alpha$ and $\beta$. This specific mixture creates a sharp spike at $\tau = 0.5$, resulting in a large derivative of the density at that point, which slightly violates Assumption (A2). Despite this, our numerical simulations under $r = 0.75$ and $\tau = 0.5$ confirm that our method remains valid. The results are shown in Figure A.8.

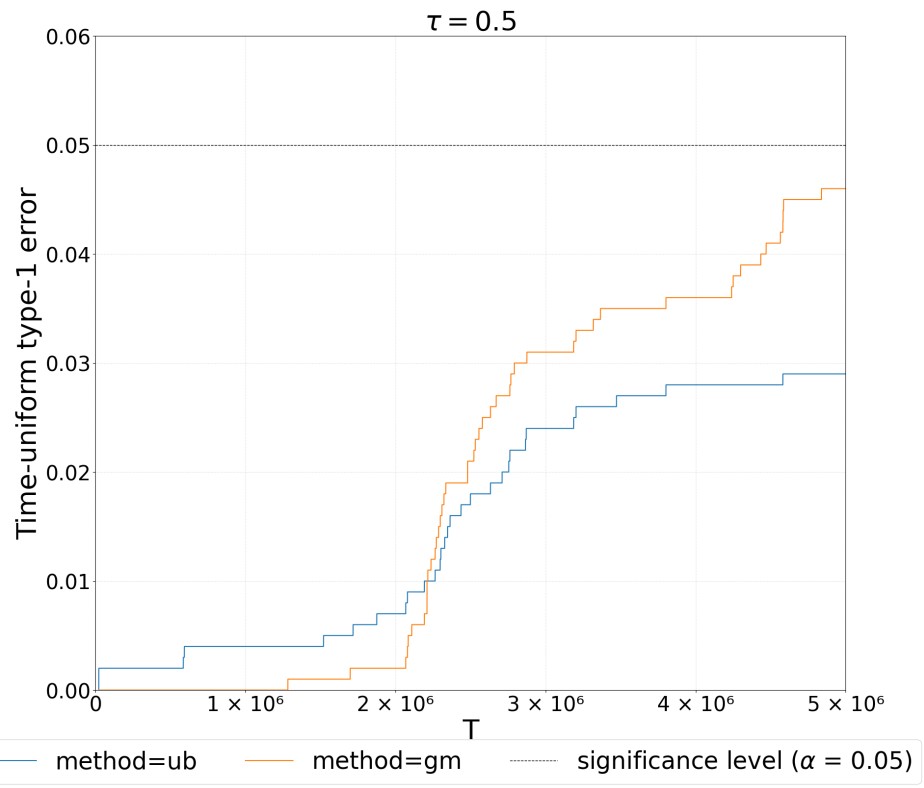

Figure A.8: Time-uniform type I error for AsypmCS constructed by by (4) and (6), when confidential data come from a mixture of Beta distributions with $\tau = 0.5$ and $r = 0.75$.

## A.2 Comparison between the proposed method and [38]

We conduct additional experiments to compare our proposed quantile estimation and confidence interval in Corollary 1 with [38]. Recall that [38] adopts a pointwise estimation approach and employs self-normalization for inference. Specifically, we considered the same simulation setting as in Section 4, with total sample size $T = 5,000,000$, quantile level $\tau = 0.3, 0.5, 0.8$, the truthful response rates $r = 0.25, 0.5, 0.75, 0.9, 1$ and distribution type set to normal. The results are in figure A.9. While both methods achieve empirical coverage rates close to the nominal confidence level, the average length of our confidence interval is more narrow across various settings of $r$ and $\tau$, indicating higher efficiency of our approach.

For point estimation accuracy of quantiles, we use the same simulation settings as in the confidence-interval study and evaluate performance by the mean squared error (MSE) of the estimated quantiles. The detailed comparison results are provided in fig A.10. We find that when $\tau$ is close to $0.5$, our method achieves comparable MSE to that in [38]. However, as $\tau$ deviates from $0.5$, the MSE of our method becomes slightly worse. This can be attributed to the dynamically chained parallel procedure used procedure employed in our quantile inference.

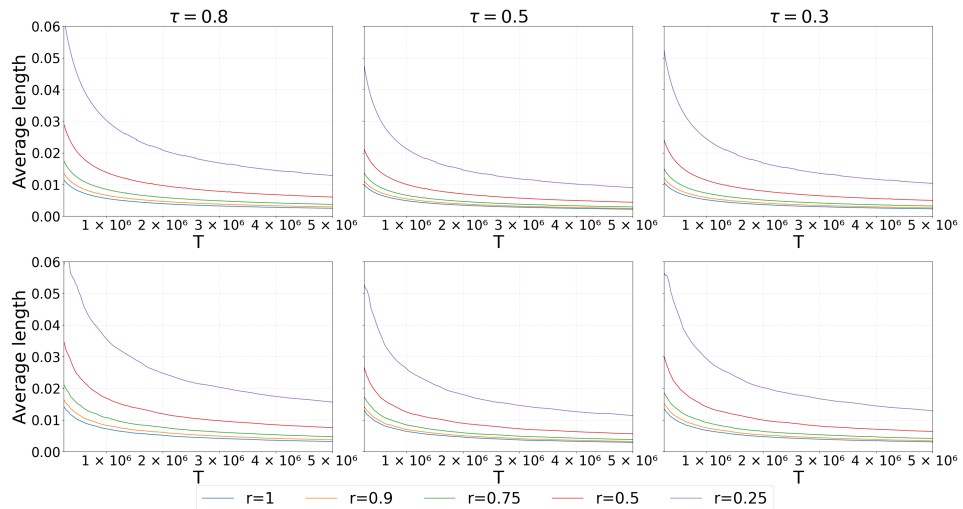

Figure A.9: Average length onstructed by the proposed method in Corollary 1 (on top panel) and [38]) (on bottom panel), when confidential data come from standard Normal $\mathcal{N}(0,1)$ with $\tau = 0.3, 0.5, 0.8$ and $r = 0.25, 0.5, 0.75, 0.9, 1$.

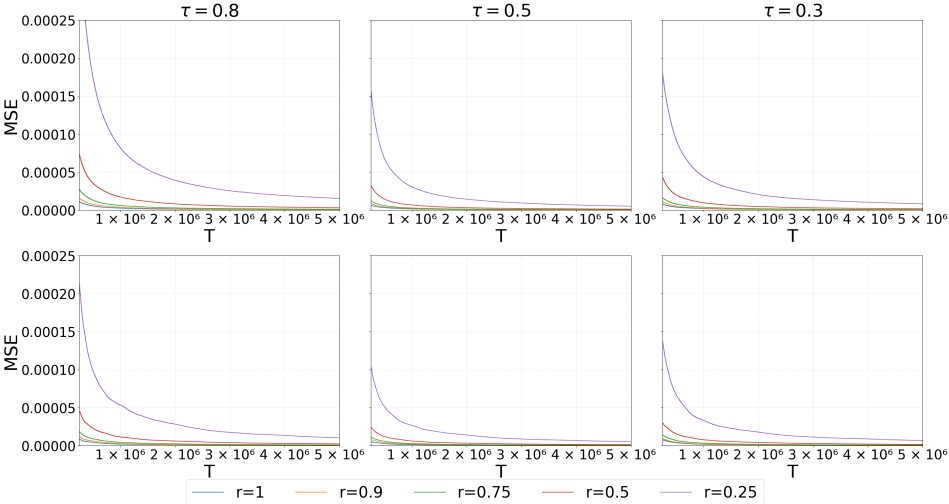

Figure A.10: MSE constructed by the proposed method in Corollary 1 (on top panel) and [38]) (on bottom panel), when confidential data come from standard Normal $\mathcal{N}(0,1)$ with $\tau = 0.3, 0.5, 0.8$ and $r = 0.25, 0.5, 0.75, 0.9, 1$.

## A.3 The selection and sensitivity analysis of tuning parameters

The proposed method requires the selection of several tuning parameters. This subsection conducts a comprehensive sensitivity analyses to show that the results are robust to variations in these choices.

Note that our tuning parameters fall into three categories. The first includes SGD-based parameters (e.g., the learning rate–related parameter $a$). Selecting the learning rate is indeed a well-known challenge in practice: SGD can be sensitive to this choice, especially in high-dimensional sparse settings and in rare–frequent or heavy-tailed regimes; see [6, 13, 19]. Nevertheless, under appropriate conditions, for example, when the objective is convex and smooth, or when the initialization is sufficiently close to the true parameter, Polyak–Ruppert averaged SGD enjoys provable convergence with tolerable sensitivity to the learning rate [43, 49, 16]. As later reported in our sensitivity studies, our results are robust to reasonable variations in this hyperparameter. The second category includes tuning parameters related to time-uniform inference, such as the AsympCS starting index $m$ and the hyperparameter $\rho$ in the Gaussian mixture bound (equation (6)). The third category consists of tuning parameters specific to our proposed method, such as the number of chains $h(t)$. We find that the results are not sensitive to these parameters; thus, recommendations from the time-uniform inference literature [54] and the default setting provided in our paper (e.g., $h(t) = \lfloor 8 \log_{10}(t) \rfloor$) can serve as practical choices.

We next conduct comprehensive sensitivity analyses for the aforementioned tuning parameters (i.e., $a$, $m$, $\rho$, and $h(t)$). Specifically, we consider one of the simulation settings from Section 4, with a total sample size of $T = 5{,}000{,}000$, $1{,}000$ repetitions, truthful response rate $r = 0.75$, quantile level $\tau = 0.5$, and normally distributed data. We evaluate the time-uniform type I error for AsympCS across a range of hyperparameter choices. The results are summarized in the following Figures A.11 to A.14. The proposed methods maintain the nominal type I error rate (5%) for nearly all hyperparameter choices, demonstrating its insensitivity to these tuning parameters.

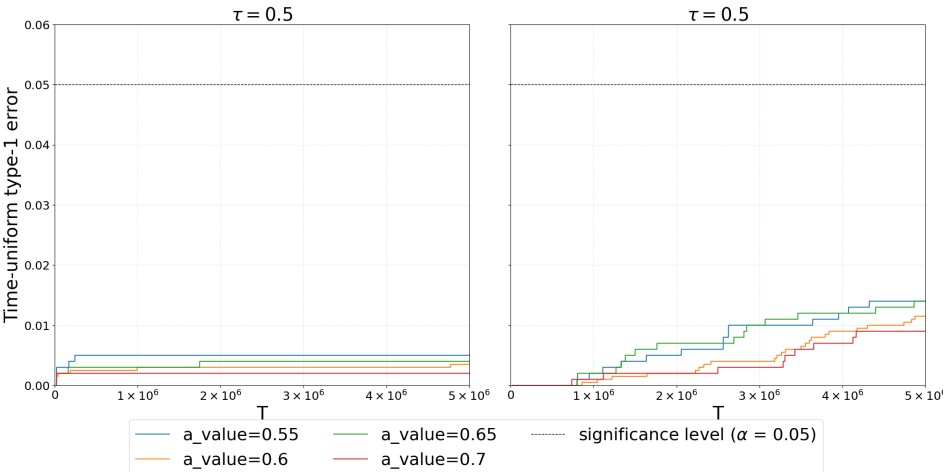

Figure A.11: Time-uniform type I error for AsypmCS constructed by (4) (on left panel) and (6) (on right panel), when confidential data come from standard Normal $\mathcal{N}(0,1)$ with $a = 0.55, 0.6, 0.65, 0.7$.

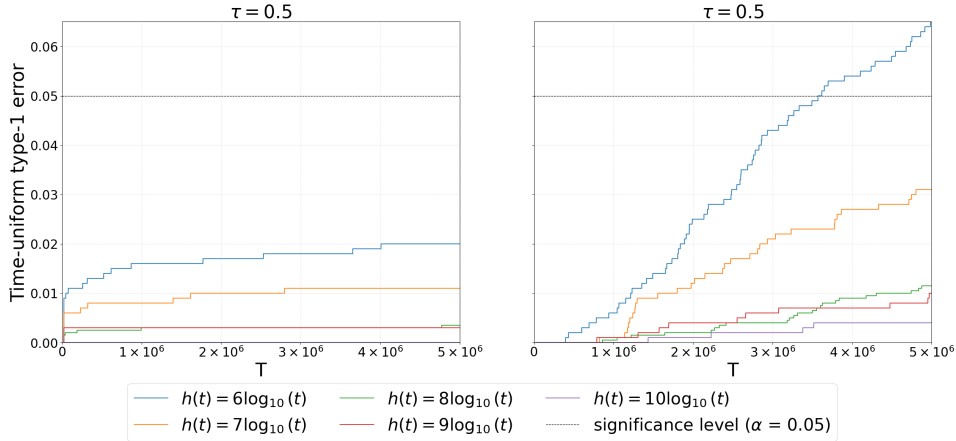

Figure A.12: Time-uniform type I error for AsypmCS constructed by (4) (on left panel) and (6) (on right panel), when confidential data come from standard Normal $\mathcal{N}(0,1)$ with $h(t) = 6\log_{10}(t), 7\log_{10}(t), 8\log_{10}(t), 9\log_{10}(t), 10\log_{10}(t)$.

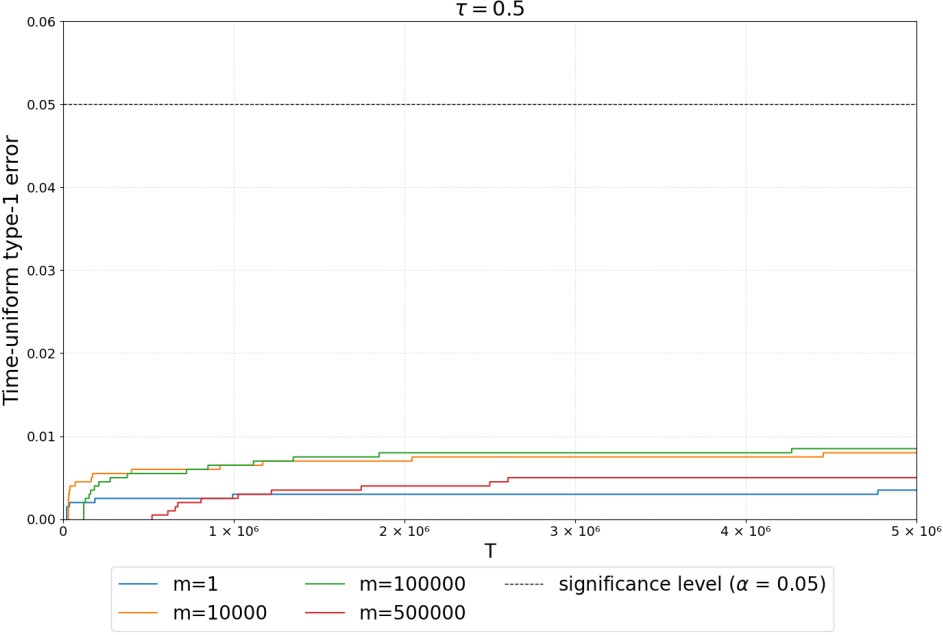

Figure A.13: Time-uniform type I error for AsypmCS constructed by (4), when confidential data come from standard Normal $\mathcal{N}(0,1)$ with $m = 1, 10000, 100000, 500000$.

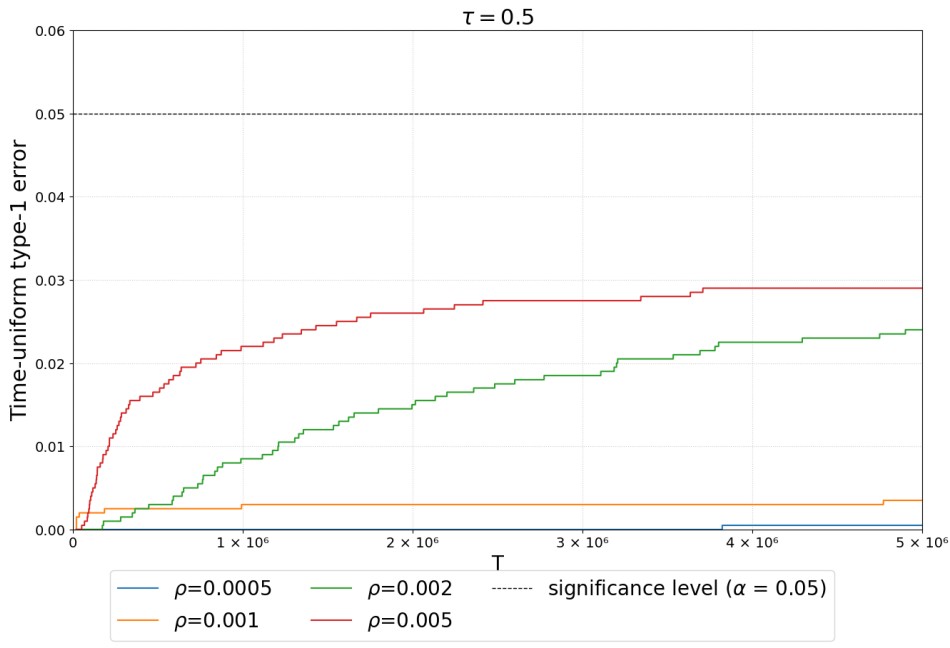

Figure A.14: Time-uniform type I error for AsypmCS constructed by (6), when confidential data come from standard Normal $\mathcal{N}(0, 1)$ with $0.0005, 0.001, 0.002, 0.005$.

## B  Proofs

This section includes detailed proofs of the theoretical results in the main article.

Elementary calculation shows that

$$g(x) := \mathbb{E}G(x, \zeta) = r\left\{F(x) - F(x^*)\right\},$$

which will be frequently used in our proofs.

**Proofs of Theorem 1**: Define the weight $\omega_k = T_k/T$. One rewrites

$$\frac{1}{T}\sum_{k=1}^{\kappa}\sum_{j=1}^{T_k}(x_{k,j} - x^*) = \sum_{k=1}^{\kappa}\frac{T_k}{T}\frac{1}{T_k}\sum_{j=1}^{T_k}(x_{k,j} - x^*) =: \sum_{k=1}^{\kappa}\omega_k\mathcal{T}_k.$$

There are two possible cases for the value of $T_k$: either $T_k \in (T/\kappa - 1, T/\kappa + 1)$ for all $1 \le k \le \kappa$ (case 1), or $T_1 = T_2 = \cdots = T_{\kappa_0} \ge T_{\kappa_0+1} = \cdots = T_{\kappa-1} \ge T_\kappa$ and $|T_{\kappa_0} - T_{\kappa_0+1}| \le 1$, $T_k \asymp T/\kappa$ for any $1 \le k \le \kappa - 1$ (case 2). Define

$$\varepsilon_{k,t} = g(x_{k,t-1}) - G(x_{k,t-1}, \zeta_{k,t}), \quad \widetilde{\varepsilon}_{k,t} = g(x^*) - G(x^*, \zeta_{k,t}).$$

Elementary calculation shows that

$$\begin{aligned}\mathbb{E}(\varepsilon_{k,t}^2|\mathcal{F}_{k,t-1}) &= \frac{1 + r - 2rF(x_{k,t-1})}{2} - \left(\frac{1 + r - 2rF(x_{k,t-1})}{2}\right)^2 \\ &= \frac{1 - r^2(2F(x_{k,t-1}) - 1)^2}{4} \\ &\xrightarrow{\mathbb{P}} \frac{1 - r^2(2\tau - 1)^2}{4},\end{aligned}$$

where the convergence in probability holds by the consistency of the quantile estimation and the continuous mapping theorem. Denote $\gamma_{k,t} = x_{k,t} - x^*$, $H = rf(x^*)$, $B_t = 1 - \eta_t H$, $A_j^t = \sum_{s=j}^{t}\left(\prod_{i=j+1}^{s}B_i\right)\eta_i$ for any $j \le t$. We decompose that

$$\begin{aligned}\mathcal{T}_k &= \frac{1}{T_k}\sum_{j=1}^{T_k}(x_{k,j} - x^*) = \frac{1}{T_k}\sum_{j=1}^{T_k}\gamma_{k,j} \\ &= \frac{1}{T_k}A_0^{T_k-1}B_0\gamma_{k,0} + \frac{1}{T_k}\sum_{j=0}^{T_k-1}A_j^{T_k-1}r_{k,j} + \frac{1}{T_k}\sum_{j=0}^{T_k-1}\left(A_j^{T_k-1} - H^{-1}\right)\varepsilon_{k,j+1} \\ &\quad + \frac{1}{T_k}\sum_{j=0}^{T_k-1}H^{-1}\left(\varepsilon_{k,j+1} - \widetilde{\varepsilon}_{k,j+1}\right) + \frac{1}{T_k}\sum_{j=0}^{T_k-1}H^{-1}\widetilde{\varepsilon}_{k,j+1} \\ &=: \mathcal{T}_{k,1} + \mathcal{T}_{k,2} + \mathcal{T}_{k,3} + \mathcal{T}_{k,4} + \mathcal{T}_{k,5},\end{aligned}$$

in which

$$r_{k,j} = H(x_{k,j} - x^*) - g(x_{k,j}).$$

**For $\mathcal{T}_{k,1}$**: According to Lemma C.4 of [54], one has $\left|A_0^{t-1}\right| \le C_0$ uniformly for all $t \ge 1$. Further observe that $\gamma_{k,0} \equiv x_0 - x^*$ for all $1 \le k \le \kappa$, thus one obtains

$$\left|\sum_{k=1}^{\kappa}\omega_k\mathcal{T}_{k,1}\right| = \mathcal{O}\left(\kappa T^{-1}\right).$$

**For $\mathcal{T}_{k,2}$**: Theorem 5 of [25] shows that

$$\max_{1 \le k \le \kappa}\mathbb{E}\left|x_{k,t} - x^*\right|^2 \lesssim \eta_t.$$

Since $|r_{k,t}| \lesssim |x_{k,t} - x^*|^2$ by Assumption (A2), we show that

$$
\sum_{t=2}^{\infty} \frac{\mathbb{E} \sum_{k=1}^{\kappa} \omega_k |r_{k,t}|}{t^{1-a} \log^{1+\epsilon}(t)} \lesssim \sum_{t=2}^{\infty} \frac{\sum_{k=1}^{\kappa} \omega_k \mathbb{E} |x_{k,t} - x^*|^2}{t^{1-a} \log^{1+\epsilon}(t)}
$$
$$
\lesssim \sum_{t=2}^{\infty} \frac{\max_{1 \leq k \leq \kappa} \mathbb{E} |x_{k,t} - x^*|^2}{t^{1-a} \log^{1+\epsilon}(t)} \tag{7}
$$
$$
\lesssim \sum_{t=2}^{\infty} \frac{1}{t \log^{1+\epsilon}(t)} < \infty.
$$

Hence, with probability one,

$$
\sum_{t=2}^{\infty} \frac{\sum_{k=1}^{\kappa} \omega_k |r_{k,t}|}{t^{1-a} \log^{1+\epsilon}(t)} < \infty.
$$

According to the uniform boundedness of $\left|A_j^{t-1}\right|$, for case 1, one shows that

$$
\left| \sum_{k=1}^{\kappa} \omega_k \mathcal{T}_{k,2} \right| \lesssim \sum_{k=1}^{\kappa} \omega_k \frac{1}{T_k} \sum_{j=0}^{T_k - 1} |r_{k,j}|
$$
$$
\leq \frac{\lfloor T/\kappa + 1 \rfloor}{\lceil T/\kappa - 1 \rceil} \frac{1}{\lfloor T/\kappa + 1 \rfloor} \sum_{j=0}^{\lfloor T/\kappa + 1 \rfloor - 1} \sum_{k=1}^{\kappa} \omega_k |r_{k,j}|
$$
$$
= \mathcal{O}_{a.s.} \left( (\kappa/T)^a \log^{1+\epsilon}(T) \right).
$$

For case 2, one shows that

$$
\left| \sum_{k=1}^{\kappa} \omega_k \mathcal{T}_{k,2} \right| \lesssim \sum_{k=1}^{\kappa} \omega_k \frac{1}{T_k} \sum_{j=0}^{T_k - 1} |r_{k,j}|
$$
$$
\leq \sum_{k=1}^{\kappa_0} \omega_k \frac{1}{T_k} \sum_{j=0}^{T_k - 1} |r_{k,j}| + \sum_{k=\kappa_0 + 1}^{\kappa - 1} \omega_k \frac{1}{T_k} \sum_{j=0}^{T_k - 1} |r_{k,j}| + \frac{1}{T} \sum_{j=0}^{T_\kappa - 1} |r_{\kappa,j}|
$$
$$
= \mathcal{O}_{a.s.} \left( (\kappa/T)^a \log^{1+\epsilon}(T) \right).
$$

**For $\mathcal{T}_{k,3}$:** For any fixed $p > 0$ (large enough), note that $\max_{1 \leq k \leq \kappa} \mathbb{E} |\varepsilon_{k,j}|^{2p} = \mathbb{E} |\varepsilon_{1,j}|^{2p}$ is bounded. Following the arguments in [54], one has $\|\mathcal{T}_{3,k}\|_{2p} = \mathcal{O}\left(T_k^{-1+a/2}\right)$. Then, using the Lemma A in Chapter 9.2.6 of [46] and the independence over $k$, one has that

$$
\left\| \sum_{k=1}^{\kappa} \omega_k \mathcal{T}_{k,3} \right\|_{2p} = \mathcal{O}\left( \kappa^{-1/2}(T/\kappa)^{-1+a/2} \right),
$$

which implies that

$$
\left| \sum_{k=1}^{\kappa} \omega_k \mathcal{T}_{k,3} \right| = \mathcal{O}_{a.s.} \left( \kappa^{-1/2}(T/\kappa)^{-1+a/2} T^{1/(2p)} \log^{1/(2p)+\epsilon}(T) \right).
$$

**For $\mathcal{T}_{k,4}$:** Observe that $\varepsilon_{k,j} - \widetilde{\varepsilon}_{k,j} = g(x_{k,j-1}) - G(x_{k,j-1}, \zeta_{k,j}) + G(x^*, \zeta_{k,j})$, and

$$
\mathbb{E} \left( \varepsilon_{k,j} - \widetilde{\varepsilon}_{k,j} \right)^2 \lesssim \mathbb{E} g^2(x_{k,j-1}) + \mathbb{E} \left\{ G(x_{k,j-1}, \zeta_{k,j}) - G(x^*, \zeta_{k,j}) \right\}^2
$$
$$
\lesssim \mathbb{E} |x_{k,j-1} - x^*|^2 + \mathbb{E} |x_{k,j-1} - x^*|
$$
$$
\lesssim \mathbb{E} |x_{k,j-1} - x^*|^2 + \left\{ \mathbb{E} |x_{k,j-1} - x^*|^2 \right\}^{1/2} \lesssim \eta_t^{1/2},
$$

where the last inequality holds by Theorem 5 of [25], and the constant does not depend on $k$.

We rewrite

$$\sum_{k=1}^{\kappa} \omega_k \mathcal{T}_{k,4} = \frac{t_T}{T} \frac{1}{t_T} \sum_{t=1}^{t_T} \sum_{k=1}^{\kappa_t} H^{-1} \left(\varepsilon_{k,t} - \widetilde{\varepsilon}_{k,t}\right),$$

where $t_T = \max_{1 \le k \le \kappa} T_k \asymp T/\kappa$ and $\kappa_t = |\{k : T_k \ge t\}| \le \kappa$. Notice that

$$\mathrm{Var}\left\{\sum_{k=1}^{\kappa_t} H^{-1}\left(\varepsilon_{k,t} - \widetilde{\varepsilon}_{k,t}\right)\right\} \lesssim \kappa_t \mathrm{Var}\left(\varepsilon_{k,t} - \widetilde{\varepsilon}_{k,t}\right) \lesssim \kappa \eta_t^{1/2}.$$

Hence,

$$\sum_{t=2}^{\infty} \mathrm{Var}\left(\frac{\sum_{k=1}^{\kappa_t} H^{-1}\left(\varepsilon_{k,t} - \widetilde{\varepsilon}_{k,t}\right)}{\kappa^{1/2} t^{1/2-a/4} \log^{1/2+\epsilon}(t)}\right) \lesssim \sum_{t=2}^{\infty} \frac{1}{t \log^{1+2\epsilon}(t)} < \infty,$$

which implies that (by Kronecker's lemma),

$$\left|\sum_{k=1}^{\kappa} \omega_k \mathcal{T}_{k,4}\right| = \mathcal{O}_{a.s.}\left(\kappa^{-1/2}(T/\kappa)^{-1/2-a/4} \log^{1/2+\epsilon}(T)\right).$$

**For $\mathcal{T}_{k,5}$:** Elementary calculation shows that

$$\mathbb{E}\widetilde{\varepsilon}_{k,j}^2 = \frac{1 - r^2(2\tau - 1)^2}{4} =: S.$$

Applying Theorem 2.6.7 of [14] with $H(x) = x^{2p}$ and $x_n = n^{\beta_0}$, there exist i.i.d. standard normal $\widetilde{Z}_{k,j}$'s and some $a, C > 0$ (depending on the distribution of $H^{-1}\widetilde{\varepsilon}_{k,j}$) such that

$$\mathbb{P}\left(\left|\sum_{k=1}^{\kappa}\sum_{t=1}^{T_k} \frac{H^{-1}\widetilde{\varepsilon}_{k,t}}{\sqrt{H^{-1}SH^{-1}}} - \sum_{i=1}^{T} \widetilde{Z}_i\right| > T^{\beta_0}\right) \le Ca^{-2p}T^{1-2p\beta_0}.$$

Thus,

$$\mathbb{P}\left(\left|\frac{1}{T}\sum_{k=1}^{\kappa}\sum_{t=1}^{T_k} \frac{H^{-1}\widetilde{\varepsilon}_{k,t}}{\sqrt{H^{-1}SH^{-1}}} - \frac{1}{T}\sum_{i=1}^{T} \widetilde{Z}_i\right| > T^{-1+\beta_0}\right) \lesssim T^{1-2p\beta_0}.$$

For $p > 2$, one selects $\beta_0 \in (1/p, 1/2)$, the Borel-Cantelli lemma leads to

$$\left|\sum_{k=1}^{\kappa} \omega_k \mathcal{T}_{k,5} - \frac{1}{T}\sum_{i=1}^{T} Z_i\right| = \mathcal{O}_{a.s.}\left(T^{-1+\beta_0}\right),$$

where $Z_i$'s are i.i.d. normal r.v.'s with mean zero and covariance $H^{-1}SH^{-1}$.

Therefore, we obtain that

$$\left|\frac{1}{T}\sum_{k=1}^{\kappa}\sum_{t=1}^{T_k}(x_{k,t} - x^*) - \frac{1}{T}\sum_{i=1}^{T} Z_i\right| = \mathcal{O}_{a.s.}\left(\sqrt{\frac{\log\log T}{T}}\right),$$

which completes the proof.

**Proofs of Theorem 2:** Recall that the weight $\omega_k = T_k/T$. We rewrite

$$\widehat{\sigma}_T^2 = \sum_{k=1}^{\kappa} \omega_k \left\{\frac{1}{\sqrt{T_k}}\sum_{j=1}^{T_k}(x_{k,j} - x^*)\right\}^2 - \left\{\sum_{k=1}^{\kappa} \omega_k \frac{1}{\sqrt{T_k}}\sum_{j=1}^{T_k}(x_{k,j} - x^*)\right\}^2.$$

Recall the definitions of $\mathcal{T}_{k,j}$, $1 \le j \le 5$.

**For $\mathcal{T}_{k,1}$:** According to Lemma C.4 of [54], one has $\left|A_0^{t-1}\right| \le C_0$ uniformly for all $t \ge 1$. Further observe that $\gamma_{k,0} \equiv x_0 - x^*$ for all $1 \le k \le \kappa$, thus one obtains $\|\mathcal{T}_{k,1}\|_2 = \mathcal{O}\left(T_k^{-1}\right)$, where the constant does not depend on $k$.

**For $\mathcal{T}_{k,2}$:** Consider that

$$\left| \frac{1}{T_k} \sum_{j=0}^{T_k-1} A_j^{T_k-1} r_{k,j} \right| \lesssim \frac{1}{T_k} \sum_{j=0}^{T_k-1} |r_{k,j}| \lesssim \frac{1}{T_k} \sum_{j=0}^{T_k-1} |x_{k,j} - x^*|^2.$$

Then,

$$\left\| \frac{1}{T_k} \sum_{j=0}^{T_k-1} A_j^{T_k-1} r_{k,j} \right\|_2 \lesssim \frac{1}{T_k} \sum_{j=0}^{T_k-1} \|x_{k,j} - x^*\|_4^2.$$

Applying Theorem 5 of [25], we have

$$\max_{1 \le k \le \kappa} \mathbb{E} |x_{k,j} - x^*|^4 \lesssim \eta_j^2.$$

Since $\eta_j \asymp j^{-a}$ with $a > 1/2$, it follows that $\|\mathcal{T}_{k,2}\|_2 = o\left(T_k^{-1/2}\right)$.

**For $\mathcal{T}_{k,3}$:** As shown in the proof of Theorem 1, one has $\|\mathcal{T}_{k,3}\|_{2p} = \mathcal{O}\left(T_k^{-1+a/2}\right) = o\left(T_k^{-1/2}\right)$, since $a < 1$.

**For $\mathcal{T}_{k,4}$:** Observe that $\sum_{j=1}^{T_k} H^{-1} \left(\varepsilon_{k,j} - \widetilde{\varepsilon}_{k,j}\right)$ is a martingale for each $k$ (independent over $1 \le k \le \kappa$), Burkholder's inequality entails that

$$\left\| \sum_{j=1}^{T_k} H^{-1} \left(\varepsilon_{k,j} - \widetilde{\varepsilon}_{k,j}\right) \right\|_2 \lesssim \left\{ \sum_{j=1}^{T_k} \left\| H^{-1} \left(\varepsilon_{k,j} - \widetilde{\varepsilon}_{k,j}\right) \right\|_2^2 \right\}^{1/2}$$

$$\lesssim \left( \sum_{j=1}^{T_k} \eta_j^{1/2} \right)^{1/2} \lesssim T_k^{\{1-a/2\}/2}.$$

Hence, for any $1 \le k \le \kappa$,

$$\|\mathcal{T}_{k,4}\|_2 = \left\| \frac{1}{T_k} \sum_{j=1}^{T_k} H^{-1} \left(\varepsilon_{k,j} - \widetilde{\varepsilon}_{k,j}\right) \right\|_2 = \mathcal{O}\left(T_k^{-1/2-a/4}\right).$$

**For $\mathcal{T}_{k,5}$:** Applying Theorem 2.6.7 of [14] with $H(x) = x^{2p}$ and $x_n = vn^{\beta_0}$, there exist i.i.d. standard normal $\widetilde{Z}_{k,j}$'s and some $a_k, C_k > 0$ (depending on the distribution of $H^{-1}\widetilde{\varepsilon}_{k,j}$) such that

$$\mathbb{P}\left( \left| \sum_{j=1}^{T_k} \frac{H^{-1}\widetilde{\varepsilon}_{k,j}}{\sqrt{H^{-1}SH^{-1}}} - \sum_{j=1}^{T_k} \widetilde{Z}_{k,j} \right| > vT_k^{\beta_0} \right) \le C_k a_k^{-2p} v^{-2p} T_k^{1-2p\beta_0}.$$

Thus,

$$\mathbb{P}\left( \left| \frac{1}{T_k} \sum_{j=1}^{T_k} \frac{H^{-1}\widetilde{\varepsilon}_{k,j}}{\sqrt{H^{-1}SH^{-1}}} - \frac{1}{T_k} \sum_{j=1}^{T_k} \widetilde{Z}_{k,j} \right| > vT_k^{-1+\beta_0} \right) \lesssim v^{-2p} T_k^{1-2p\beta_0}.$$

Since $\mathbb{E}X^p = \int_0^\infty pv^{p-1}\mathbb{P}(|X| > v)dv$, we also have

$$\left\| \mathcal{T}_{k,5} - \frac{1}{T_k} \sum_{j=1}^{T_k} Z_{k,j} \right\|_2 = \mathcal{O}\left(T_k^{-1+\beta_0}\right).$$

According to the above results, we show that

$$\left\| \frac{1}{T_k} \sum_{j=1}^{T_k} (x_{k,j} - x^*) - \frac{1}{T_k} \sum_{j=1}^{T_k} Z_{k,j} \right\|_2 = o\left(T_k^{-1/2}\right),$$

which implies

$$\left\| \frac{1}{\sqrt{T_k}} \sum_{j=1}^{T_k} (x_{k,j} - x^*) \right\|_2 = \left\| \frac{1}{\sqrt{T_k}} \sum_{j=1}^{T_k} Z_{k,j} \right\|_2 + \mathcal{O}(1) < \infty,$$

$$\mathbb{E} \frac{1}{\sqrt{T_k}} \sum_{j=1}^{T_k} (x_{k,j} - x^*) = \mathbb{E} \frac{1}{\sqrt{T_k}} \sum_{j=1}^{T_k} Z_{k,j} + \mathcal{O}(1) = o(1).$$

The SLLN (independent but not identically distributed) further yields that

$$\sum_{k=1}^{\kappa} \omega_k \frac{1}{\sqrt{T_k}} \sum_{j=1}^{T_k} (x_{k,j} - x^*) \xrightarrow{a.s.} 0.$$

It sufficed to show

$$\sum_{k=1}^{\kappa} \omega_k \left\{ \frac{1}{\sqrt{T_k}} \sum_{j=1}^{T_k} (x_{k,j} - x^*) \right\}^2 \xrightarrow{a.s.} \mathbb{E} Z_{k,j}^2 = \frac{1 - r^2(2\tau - 1)^2}{4r^2 f^2(x^*)}.$$

For case 1, $T_1 = T_2 = \cdots = T_{\kappa_0} = T_{\kappa_0} + 1 = \cdots = T_\kappa + 1$. The SLLN (i.i.d.) implies that

$$\sum_{k=1}^{\kappa_0} \omega_k \left\{ \frac{1}{\sqrt{T_k}} \sum_{j=1}^{T_k} (x_{k,j} - x^*) \right\}^2 \xrightarrow{a.s.} \sum_{k=1}^{\kappa_0} \frac{T_k}{T} \mathbb{E} Z_{k,j}^2 = \sum_{k=1}^{\kappa_0} \frac{T_k}{T} \frac{1 - r^2(2\tau - 1)^2}{4r^2 f^2(x^*)},$$

$$\sum_{k=\kappa_0+1}^{\kappa} \omega_k \left\{ \frac{1}{\sqrt{T_k}} \sum_{j=1}^{T_k} (x_{k,j} - x^*) \right\}^2 \xrightarrow{a.s.} \sum_{k=\kappa_0+1}^{\kappa} \frac{T_k}{T} \mathbb{E} Z_{k,j}^2 = \sum_{k=\kappa_0+1}^{\kappa} \frac{T_k}{T} \frac{1 - r^2(2\tau - 1)^2}{4r^2 f^2(x^*)}.$$

The result is obtained by adding the above two expressions.

For case 2, the SLLN (i.i.d.) entails that

$$\sum_{k=1}^{\kappa-1} \omega_k \left\{ \frac{1}{\sqrt{T_k}} \sum_{j=1}^{T_k} (x_{k,j} - x^*) \right\}^2 - \sum_{k=1}^{\kappa-1} \omega_k \mathbb{E} Z_{k,j}^2$$

$$= \sum_{k=1}^{\kappa-1} \omega_k \left\{ \frac{1}{\sqrt{T_k}} \sum_{j=1}^{T_k} (x_{k,j} - x^*) \right\}^2 - \left(1 - \frac{T_\kappa}{T}\right) \mathbb{E} Z_{k,j}^2 \xrightarrow{a.s.} 0,$$

As $T_\kappa/T = o(1)$, it completes the proof of the consistency of $\hat{\sigma}^2$.

**Proofs of Theorem 3:** According to the law of iterated logarithm, the rate of the bound of any confidence sequence for the unknown mean of Gaussian random variables with unit variance is at least $\sqrt{T^{-1} \log\log T}$. On the one hand, Theorem 1 shows that Conditions G-1 and G-3 in [52] are satisfied. On the other hand, Theorem 2 ensures Condition G-4 in [52]. Hence, we apply Theorem 2.4 in [52] to complete the proof.

