# OpenReview forum: "Time-uniform and Asymptotic Confidence Sequence of Quantile under Local Differential Privacy"
_NeurIPS.cc/2025/Conference — NeurIPS 2025 poster_

### Official Review · Reviewer_PTJr · 2025-06-21

**Clarity:** 3
**Significance:** 3
**Originality:** 4
**Rating:** 4
**Confidence:** 3

**Summary:**

The paper is the first to extend the asymptotic confidence sequences of the quantile estimation framework introduced by Waudby-Smith et al. [41] to differential privacy.  It uses a parallel SGD method that was recently applied to high-confidence interval problems by Wanrong Zhu et al. [46]. Differential privacy is ensured through a randomized response mechanism used with gradient-like updates proposed by Liu et al. [32].

**Questions:**

1) The paper states that the proposed approximation rate is faster than that given by the Law of the Iterated Logarithm (LIL). However, isn't the LIL rate of order $ O\left(\frac{1}{\sqrt{t \log \log t}}\right)$, which appears faster than the rate you claim?

2) Would high-confidence intervals alone suffice for private quantile estimation with confidence intervals? What advantages does the asymptotic sequence framework offer in practice?

3) If the goal was only quantile estimation without confidence intervals, how would your procedure compare to that of Liu et al. [32]?

**Ethical Concerns:**

["NO or VERY MINOR ethics concerns only"]

**Final Justification:**

This paper is the first to bring the time uniform quantile estimation with confidence intervals problem into differential privacy. In doing so, it generalizes previous non-private works. I find the paper to be interesting, and the technical contributions are solid and non-trivial.

In the rebuttal, the authors provided additional justification for the practicality of the asymptotic sequence framework. While I find their arguments reasonable, they are not fully convincing; therefore, I maintain my borderline accept score.

My concerns regarding the paper’s presentation are promised to be resolved.

**Limitations:**

yes

**Paper Formatting Concerns:**

I do not have any concerns

**Quality:**

3

**Strengths And Weaknesses:**

This paper is the first to bring the time uniform quantile estimation with confidence intervals problem into differential privacy. In doing so, it generalizes previous non-private works. I find the paper to be interesting, and the technical contributions are solid and non-trivial.

However, I am not fully convinced that the asymptotic sequence framework is the most appropriate choice for practical applications. Would high-confidence intervals alone be sufficient?

Additionally, the paper would benefit from further polishing:

Presentation : The fonts in all of the plots are too small and difficult to read.


There are issues with spacing and punctuation in citations, such as: “Privacy , see [16]”, “Privacy,see [26]”, “(e.g., 7)”

---

> ### Author Rebuttal · Authors · 2025-07-30
>
> We sincerely appreciate the time and effort you've invested in reviewing our paper. Your feedback is invaluable to us, and we have addressed each of your concerns below. Should you have any further questions, please do not hesitate to ask.
>
>
> **For weakness:**
> W1:
> In practice, it has become standard for organizations with an online presence to conduct large-scale A/B testing [1,2] to improve product performance and user experience. These experiments are inherently sequential, with users arriving over time and results observed quickly.
> Unlike clinical trials or industrial quality control, where formal planning is the norm, online experiments are often run with little predefined stopping rules and rely on dynamic decision-making. In such settings, confidence sequences offer a principled alternative, providing valid inference at arbitrary stopping times without sacrificing statistical rigor.
>
> While classical confidence intervals are indeed sufficient in fixed-sample settings, the asymptotic confidence sequence framework is particularly well-suited for modern practical applications that involve continuous monitoring and sequential decision-making. Otherwise, using confidence intervals may lead to severely uncontrolled type-I error, as evidenced in [3,4]. Thus, our use of the asymptotic CS framework is highly  practically motivated.
>
> When the number of chains $K$ is fixed, $\widehat{\sigma}_T^2 $ is no longer a consistent estimator of $\sigma^2$. According to  the proof of Theorem 1, we can derive that the statistic $\sqrt{T}\widehat\sigma_T^{-1}\widehat x_T$ asymptotically follows a $t$-distribution with $K-1$ degrees of freedom, which entails that an  asymptotic confidence interval for the quantile at time $T$ takes the form:
>
> $ [\widehat x_T - \widehat\sigma_T t_{K-1,1-\alpha/2} / \sqrt{T},   $ $\widehat x_T + \widehat\sigma_T t_{K-1,1-\alpha/2} / \sqrt{T}]$
>
>
>
> similar to the  high-confidence interval proposed in [5]. This indeed improves the accuracy of the confidence interval, as it  holds for any fixed number of chains $K$. However, constructing confidence sequences based on the
> $t$-distribution is highly challenging, as existing results in the literature predominantly focus on sub-Gaussian or sub-Gamma random variables; see [6].
>
>
> W2: "Additionally, the paper would benefit from further polishing."
>
> We have carefully revised the manuscript to improve the clarity, organization, and overall presentation.
>
> W3: "Presentation : The fonts in all of the plots are too small and difficult to read."
>
> We have carefully revised all plots by increasing the font sizes to ensure that the figures are clearer and more readable.
>
> W4: "here are issues with spacing and punctuation in citations, such as: “Privacy , see [16]”, “Privacy,see [26]”, “(e.g., 7)"
>
> We have corrected these citation issues and carefully reviewed the entire manuscript to ensure that all spacing and punctuation in citations are now properly formatted.
>
> **For questions**
>
>
> Q1: For a sequence of i.i.d. random variables $\{X_i\}$ with mean $0$ and variance $\sigma^2$, let $S_n = \sum_{i=1}^n X_i$. The Law of the Iterated Logarithm (LIL) states:
>
> $$\limsup_{n \to \infty} \frac{S_n}{\sqrt{2n \log \log n}} = \sigma \quad \text{almost surely}.$$
>
>
> Thus, for sample mean $S_n/n$, the LIL rate is of order $O_{a.s.}\left(\sqrt{ \log \log n/n}\right)$. Note that the strong approximation rate established in Theorem 1 is of order
> $o_{a.s.}\left(\sqrt{ \log \log n/n}\right)$, which is strictly faster than the rate given by the law of the iterated logarithm.
>
>
> Q2: Kindly refer to weakness 1.
>
> Q3: we conducted additional experiments to directly compare our method with that of [7], focusing solely on point estimation accuracy for quantiles. Specifically, we considered the same simulation setting as in Section 4, with total sample size $T = 5,000,000$, quantile level $\tau = 0.3,0.5,0.8$, the truthful response rates $r=0.25,0.5,0.75,0.9,1$ and distribution type set to normal. The evaluation metric is the mean squared error (MSE) of the estimated quantiles.  The detailed comparison results are provided in Tables 1-5. We find that when $\tau$ is close to $0.5$, our method achieves comparable MSE to that in [7]. However, as $\tau$ deviates from $0.5$, the MSE of our method becomes slightly worse. This can be attributed to the  dynamically chained parallel procedure used procedure employed in our quantile inference.
>
> **Table 1.** MSE ($\times 10^{-7}$) for $T = 5 \times 10^6$.
>
> | method | $\tau$ | $r = 1$   | $r = 0.9$  | $r = 0.75$ | $r = 0.5$  | $r = 0.25$ |
> |--------|--------|-----------|------------|------------|------------|------------|
> |        |        |           |   MSE      | ($\times 10^{-7}$) |   |   |
> | sn     | 0.8    | 4.49      | 5.71       | 9.53       | 22.7       | 105        |
> |        | 0.5    | 3.22      | 3.80       | 5.57       | 12.6       | 53.6       |
> |        | 0.3    | 3.75      | 4.63       | 6.99       | 16.2       | 70.6       |
> | Our   | 0.8    | 5.47      | 7.48       | 12.2       | 31.8       | 154        |
> |        | 0.5    | 3.20      | 4.04       | 5.71       | 13.1       | 52.5       |
> |        | 0.3    | 4.11      | 4.94       | 7.43       | 18.3       | 82.3       |
>
>
> **Table 2.** MSE ($\times 10^{-7}$) for $T = 4 \times 10^6$.
>
> | method | $\tau$ | $r = 1$   | $r = 0.9$  | $r = 0.75$ | $r = 0.5$  | $r = 0.25$ |
> |--------|--------|-----------|------------|------------|------------|------------|
> |        |        |           |   MSE      | ($\times 10^{-7}$) |   |   |
> | SN-based     | 0.8    | 5.61      | 7.25       | 11.4       | 29.2       | 128        |
> |        | 0.5    | 4.13      | 4.84       | 6.78       | 16.0       | 61.1       |
> |        | 0.3    | 4.52      | 5.49       | 8.53       | 21.1       | 80.7       |
> | Our   | 0.8    | 6.68      | 9.23       | 15.4       | 39.7       | 193        |
> |        | 0.5    | 3.98      | 5.21       | 7.02       | 16.5       | 67.7       |
> |        | 0.3    | 4.98      | 6.23       | 9.30       | 23.4       | 104        |
>
> **Table 3.** MSE ($\times 10^{-7}$) for $T = 3 \times 10^6$.
> | method | $\tau$ | $r = 1$   | $r = 0.9$  | $r = 0.75$ | $r = 0.5$  | $r = 0.25$ |
> |--------|--------|-----------|------------|------------|------------|------------|
> |        |        |           |   MSE      | ($\times 10^{-7}$) |   |   |
> | SN-based     | 0.8    | 7.26      | 9.61       | 15.3       | 40.0       | 176        |
> |        | 0.5    | 5.66      | 6.74       | 9.04       | 21.9       | 83.9       |
> |        | 0.3    | 6.19      | 7.46       | 11.3       | 29.6       | 110        |
> | Our   | 0.8    | 8.84      | 12.5       | 20.2       | 53.5       | 255        |
> |        | 0.5    | 5.25      | 6.67       | 9.43       | 22.0       | 94.5       |
> |        | 0.3    | 6.66      | 8.32       | 12.8       | 30.6       | 138        |
>
> **Table 4.** MSE ($\times 10^{-7}$) for $T = 2 \times 10^6$.
>
> | method | $\tau$ | $r = 1$   | $r = 0.9$  | $r = 0.75$ | $r = 0.5$  | $r = 0.25$ |
> |--------|--------|-----------|------------|------------|------------|------------|
> |        |        |           |   MSE      | ($\times 10^{-7}$) |   |   |
> | SN-based     | 0.8    | 10.5      | 13.9       | 21.8       | 60.6       | 256        |
> |        | 0.5    | 8.25      | 10.2       | 13.0       | 32.5       | 126        |
> |        | 0.3    | 9.58      | 12.1       | 16.7       | 43.5       | 166        |
> | Our   | 0.8    | 13.0      | 19.2       | 30.0       | 82.8       | 394        |
> |        | 0.5    | 7.64      | 10.0       | 14.1       | 32.3       | 144        |
> |        | 0.3    | 10.2      | 12.8       | 19.4       | 47.4       | 207        |
>
> **Table 5.** MSE ($\times 10^{-7}$) for $T = 1 \times 10^6$.
>
> | method | $\tau$ | $r = 1$   | $r = 0.9$  | $r = 0.75$ | $r = 0.5$  | $r = 0.25$ |
> |--------|--------|-----------|------------|------------|------------|------------|
> |        |        |           |   MSE      | ($\times 10^{-7}$) |   |   |
> | SN-based     | 0.8    | 20.6      | 27.6       | 44.9       | 124        | 498        |
> |        | 0.5    | 16.1      | 19.9       | 26.4       | 64.5       | 245        |
> |        | 0.3    | 17.6      | 23.2       | 31.7       | 84.3       | 314        |
> | Our   | 0.8    | 26.6      | 38.9       | 62.3       | 174        | 823        |
> |        | 0.5    | 16.1      | 21.3       | 28.5       | 68.0       | 306        |
> |        | 0.3    | 20.7      | 25.9       | 38.6       | 96.3       | 440        |
>
>
> **Reference**
>
> [1]  Nicholas Larsen, Jonathan Stallrich, Srijan Sengupta, Alex Deng, Ron Kohavi, and Nathaniel T Stevens. Statistical challenges in online controlled experiments: A review of a/b testing methodology. The American Statistician, 78(2):135–149, 2024.
>
> [2] Dan Siroker and Pete Koomen. A/B testing: The most powerful way to turn clicks into customers. John Wiley \& Sons, 2015.
>
> [3] Peter Armitage, CK McPherson, and BC Rowe. Repeated significance tests on accumulating data. Journal of the Royal Statistical Society: Series A (General), 132(2):235–244, 1969.
>
> [4]  Ron Berman, Leonid Pekelis, Aisling Scott, and Christophe Van den Bulte. Phacking and false discovery in A/B testing, volume 10. SSRN, 2018.
>
> [5] Wanrong Zhu, Zhipeng Lou, Ziyang Wei, and Wei Biao Wu. High confidence level inference is almost free using parallel stochastic optimization. arXiv preprint arXiv:2401.09346, 2024.
>
>
> [6] Steven R Howard, Aaditya Ramdas, Jon McAuliffe, and Jasjeet Sekhon. Time-uniform, nonparametric, nonasymptotic confidence sequences. The Annals of Statistics, 49(2):1055–1080, 2021.
>
> [7] Liu, Y., Hu, Q., Ding, L., \& Kong, L. (2023, July). Online local differential private quantile inference via self-normalization. In International Conference on Machine Learning (pp. 21698-21714). PMLR.

---

> > ### Comment · Reviewer_PTJr · 2025-08-02
> >
> > Thank you for your clear and detailed response!

---

> > > ### Author Response · Authors · 2025-08-06
> > >
> > > Thank you for your encouraging feedback. We truly appreciate the time and expertise you have devoted to reviewing our paper, and we have carefully incorporated your suggestions into the revision.  If our revisions have resolved your concerns, we would be sincerely grateful if you might consider updating your rating.

---

### Official Review · Reviewer_2z3U · 2025-06-30

**Clarity:** 2
**Significance:** 3
**Originality:** 3
**Rating:** 5
**Confidence:** 2

**Summary:**

Accurately estimating the quantiles under LDP is challenging because it needs asymptotic variance estimates of the estimator. For online quantile inference (LDP reports arriving in sequence), [46] proposed a parallel SGD based framework with high confidence variance estimation. However, their variance estimator does not obey consistency due to fixed length of the chains.
This work extends the idea of using P-SGD, and proposes a new low resource online method for estimating confidence sequences for quantiles, and an almost surely consistent estimator  of the quantile variance (derived from P-SGD iterates) under LDP.
They also show that Gaussian approximation for the proposed estimator provides asymptotically anytime-valid confidence sequences with lengths  following the LIL.

**Questions:**

1)	The setting does not seem to assume any bound on the input? Please clarify.
2)	Please explain the equation on line 128 (at least in the Appendix). It’s not immediate how G(x, \Xi) can be viewed as a permuted stochastic gradient.
3)	Please discuss the continual observations model (or the differences from it) in the literature review, placing your solution in perspective. Some generic background on DP and LDP can be reduced in Section 1.
4)	Markers and legends in Figures are barely visible. Please increase the sizes.

**Ethical Concerns:**

["NO or VERY MINOR ethics concerns only"]

**Final Justification:**

Authors have promised to add missing descriptions, and added new experiments in the rebuttal.

**Limitations:**

yes

**Quality:**

3

**Strengths And Weaknesses:**

Strengths:
1)	Online quantile and its consistent variance estimation with asymptotic confidence sequences under LDP seems unsolved.
2)	The method seems easy to implement and practical.

Weakness:
1)	SGD based methods can be sensitive to the choice of hyperparameters and require tuning.

2)	The continual observation model under DP (https://dl.acm.org/doi/10.1145/1806689.1806787) is well-established in the central setting. Authors do not discuss this model in the literature review.  There are some works (e.g. https://openreview.net/attachment?id=pWeQdceMHL&name=pdf) also in the LDP setting, possibly not on the same problem though.

---

> ### Author Rebuttal · Authors · 2025-07-30
>
> We sincerely appreciate the time and effort you've invested in reviewing our paper. Your feedback is invaluable to us, and we have addressed each of your concerns below. Should you have any further questions, please do not hesitate to ask.
>
>
> **For weakness:**
>
> W1: We totally agree with you that SGD-based methods can be sensitive to the choice of hyperparameters and
> require tuning, especially in high-dimensional sparse settings and in rare–frequent or heavy-tailed regimes; see [1,2,3].
>
> We will reiterate this limitation in the revised manuscript and plan to investigate additional techniques for reducing our method's sensitivity to SGD-based parameters in future work. Nevertheless, under appropriate conditions, for example, when the objective is convex and smooth, or when the initialization is sufficiently close to the true parameter, Polyak–Ruppert averaged SGD enjoys provable convergence with tolerable sensitivity to the learning rate [4,5,6].
>
>
> Furthermore, according to your valuable comment, we have now conducted comprehensive sensitivity analyses. Specifically, we varied the learning rate hyperparameter $a$. We consider one of the simulation settings from Section 4, with a total sample size of $T = 5{,}000{,}000$, $1,000$ repetitions, truthful response rate $r = 0.75$, quantile level $\tau = 0.5$, and normally distributed data. We evaluate the time-uniform type I error for AsympCS across a range of hyperparameter choices. The results are summarized in the following Tables. We are pleased to report that, the proposed methods maintain the nominal type I error rate (5\%) for all hyperparameter choices, demonstrating its insensitivity to these tuning parameters.
>
> **Sensitivity analysis for $a$**: (t $\times 10^4$)
> | $a$ | method | t = 5 | t = 100 | t = 200 | t = 300 | t = 400 | t = 500 |
> |---|---------------|-----------|---------------|---------------|---------------|---------------|---------------|
> | 0.55 | UB | 0.003 | 0.005 | 0.005 | 0.005 | 0.005 | 0.005 |
> | 0.6 | UB | 0.002 | 0.003 | 0.003 | 0.003 | 0.003 | 0.004 |
> | 0.65 | UB | 0.002 | 0.003 | 0.004 | 0.004 | 0.004 | 0.004 |
> | 0.7 | UB | 0.002 | 0.002 | 0.002 | 0.002 | 0.002 | 0.002 |
> | 0.55 | GM | 0 | 0.002 | 0.005 | 0.01 | 0.012 | 0.014 |
> | 0.6 | GM | 0 | 0.001 | 0.002 | 0.004 | 0.009 | 0.012 |
> | 0.65 | GM | 0 | 0.002 | 0.007 | 0.01 | 0.012 | 0.014 |
> | 0.7 | GM | 0 | 0.001 | 0.002 | 0.003 | 0.007 | 0.009 |
>
>
> **Sensitivity analysis for $h(t)$** (t $\times 10^4$)
> | $h(t)$ | method | t = 5 | t = 100 | t = 200 | t = 300 | t = 400 | t = 500 |
> |---|---------------|-----------|---------------|---------------|---------------|---------------|---------------|
> | $6\log_{10}(t/5)$ | UB | 0.01| 0.016 | 0.017 | 0.018 | 0.019 | 0.02 |
> | $7\log_{10}(t/5)$ | UB | 0.006 | 0.008 | 0.01 | 0.011 | 0.011 | 0.011 |
> | $8\log_{10}(t/5)$ | UB | 0.002 | 0.003 | 0.003 | 0.003 | 0.003 | 0.004 |
> | $9\log_{10}(t/5)$ | UB | 0.003 | 0.003 | 0.003 | 0.003 | 0.003 | 0.003 |
> | $10\log_{10}(t/5)$ | UB | 0 | 0 | 0 | 0 | 0 | 0 |
> | $6\log_{10}(t/5)$ | GM | 0 | 0.006 | 0.025 | 0.043 | 0.054 | 0.065 |
> | $7\log_{10}(t/5)$ | GM | 0 | 0.001 | 0.012 | 0.021 | 0.027 | 0.031 |
> | $8\log_{10}(t/5)$ | GM | 0 | 0.001 | 0.002 | 0.004 | 0.009 | 0.012 |
> | $9\log_{10}(t/5)$ | GM | 0 | 0.001 | 0.004 | 0.006 | 0.007 | 0.01 |
> | $10\log_{10}(t/5)$ | GM | 0 | 0 | 0.001 | 0.002 | 0.004 | 0.004 |
>
>
> **Sensitivity analysis for $m$** (t $\times 10^4$)
> | $m$ | method |t = 100 | t = 200 | t = 300 | t = 400 | t = 500 |
> |---|---------------|---------------|---------------|---------------|---------------|---------------|
> | 1 | UB | 0.003 | 0.003 | 0.003 | 0.003 | 0.0035 |
> | 10000 | UB | 0.0065 | 0.007 | 0.0075 | 0.0075 | 0.008 |
> | 100000 | UB | 0.0065 | 0.008 | 0.008 | 0.008 | 0.0085 |
> | 500000 | UB | 0.0025 | 0.004 | 0.005 | 0.005 | 0.005 |
>
>
> **Sensitivity analysis for $\rho$** (t $\times 10^4$)
> | $\rho$ | method |t = 100 | t = 200 | t = 300 | t = 400 | t = 500 |
> |---|---------------|---------------|---------------|---------------|---------------|---------------|
> | 0.0005 | GM | 0 | 0 | 0 | 0 | 0.0005 | 0.0005 |
> | 0.001 | GM | 0.002 | 0.003 | 0.003 | 0.003 | 0.003 | 0.0035 |
> | 0.002 | GM | 0 | 0.0085 | 0.015 | 0.0185 | 0.0225 | 0.024 |
> | 0.005 | GM | 0 | 0.022 | 0.026 | 0.0275 | 0.029 | 0.029 |
>
>
> W2:  Thank you for bringing the interesting work [7] to our attention. Our setting, which combines sequential inference with online algorithms, is designed for scenarios involving continual observation. Specifically, we consider user-level privacy protection under LDP, where the true data cannot be accessed directly. In contrast, under continual observation or in the simpler central DP setting, one may have access to the raw data and apply privacy mechanisms in subsequent steps.
>
> In such cases, quantile inference, or even point estimation, is nontrivial; see recent developments in [8] and [9]. To achieve differential privacy under continual observation, one may compress or recompute the induced noise at each step to improve efficiency, as discussed in [10]. We will incorporate a more detailed literature review on continual observation in the revised version.
>
>
>
>
> **For questions**
>
> Q1:  Indeed, our algorithm does not require the input random variables to be bounded.
> This contrasts with classical linear statistics, such as those in [11], which require a boundedness assumption. In our approach, the quantile is estimated via SGD, leveraging the fact that the gradient is an indicator function that always outputs binary values, which allows us to avoid imposing boundedness assumptions under LDP.
>
> This is supported both theoretically and empirically. Theoretically, the proposed asymptotic confidence sequence is   justified by the Gaussian approximation established in Theorem 1, which, similar to the Gaussian approximation used in [12], does not rely on any boundedness condition. Empirically, both the Gaussian and Cauchy distributions used in our simulations are unbounded random variables, and our method performs well even under the heavy-tailed Cauchy distribution.
>
> Q2: We apologize for any confusion. The permuted input gradient is modified using the algorithm proposed in [14]. Regarding the required data, the true data is observed with probability $r$, and a Bernoulli(0.5) sample is used with probability $1 - r$. As a result, the quantile being estimated from the permuted data corresponds to $r\tau/2 + (1 - r)/2$, as stated in the equation on line 128. We will include a more detailed explanation of the LDP mechanism in the Appendix in the revised version.
>
> Q3: We will reduce the general background on DP and LDP in Section 1 and include a more detailed literature review on continual observation in the revised version.
>
> Q4: We have carefully revised all figures by enlarging the markers, and font sizes of the legends, so that the visual presentation is now clearer and easier to read.
>
>
> **Reference**
>
>
> [1] Léon Bottou, Frank E Curtis, and Jorge Nocedal. Optimization methods for large-scale machine learning. SIAM review, 60(2):223–311, 2018.
>
> [2] Jeremy Cohen, Simran Kaur, Yuanzhi Li, J Zico Kolter, and Ameet Talwalkar. Gradient descent on neural networks typically occurs at the edge of stability. In International Conference on Learning Representations, 2021.
>
> [3] John Duchi, Elad Hazan, and Yoram Singer. Adaptive subgradient methods for online learning and stochastic optimization. Journal of machine learning research, 12(7), 2011.
>
> [4] Boris T Polyak and Anatoli B Juditsky. Acceleration of stochastic approximation by averaging. SIAM journal on control and optimization, 30(4):838–855, 1992.
>
> [5]  Ilya Sutskever, James Martens, George Dahl, and Geoffrey Hinton. On the importance of initialization and momentum in deep learning. In International conference on machine learning, pages 1139–1147. pmlr, 2013.
>
>
> [6] Aymeric Dieuleveut and Francis Bach. Nonparametric stochastic approximation with large step-sizes. 2016.
>
> [7] Dwork, C., Naor, M., Pitassi, T., \& Rothblum, G. N. (2010, June). Differential privacy under continual observation. In Proceedings of the forty-second ACM symposium on Theory of computing (pp. 715-724).
>
> [8] Gillenwater, J., Joseph, M., & Kulesza, A. (2021, July). Differentially private quantiles. In International Conference on Machine Learning (pp. 3713-3722). PMLR.
>
> [9] Lalanne, C., Garivier, A., & Gribonval, R. (2023, July). Private statistical estimation of many quantiles. In International Conference on Machine Learning (pp. 18399-18418). PMLR.
>
> [10] Tenenbaum, J., Kaplan, H., Mansour, Y., & Stemmer, U. (2023, July). Concurrent shuffle differential privacy under continual observation. In International Conference on Machine Learning (pp. 33961-33982). PMLR.
>
> [11]  Ian Waudby-Smith, Steven Wu, and Aaditya Ramdas. Nonparametric extensions of randomized response for private confidence sets. In International Conference on Machine Learning, pages 36748–36789. PMLR, 2023.
>
>
> [12] Chuhan Xie, Kaicheng Jin, Jiadong Liang, and Zhihua Zhang. Asymptotic time-
> uniform inference for parameters in averaged stochastic approximation. arXiv
> preprint arXiv:2410.15057, 2024.
>
> [13] Liu, Y., Hu, Q., Ding, L., \& Kong, L. (2023, July). Online local differential private quantile inference via self-normalization. In International Conference on Machine Learning (pp. 21698-21714). PMLR.

---

> > ### Comment · Reviewer_2z3U · 2025-08-05
> >
> > Thanks for your detailed response. My concerns are answered, please include the experiments  and add the descriptions promised in the updated version. I am increasing my score.

---

> > > ### Author Response · Authors · 2025-08-06
> > >
> > > Thank you very much for your encouraging feedback and for raising our score. We are pleased that our explanations and additional experiments have addressed your concerns. We will incorporate these results and clarifications into the revision.

---

### Official Review · Reviewer_Nwru · 2025-07-02

**Clarity:** 3
**Significance:** 3
**Originality:** 3
**Rating:** 5
**Confidence:** 3

**Summary:**

This work studies quantile estimation (and its variance) in the online setting under local differential privacy. The main techniques used in the paper is a parallel stochastic gradient descent (P-SGD) which partitions the data and uses multiple parallel SGD optimizations to estimate the quantile. One of the main contributions of the work is increasing the number of partitions as the number of samples increases dynamically. The algorithm is also able to measure the variance of the estimator using the sample variance from the multiple chains. To achieve privacy, the algorithm applies randomized response on the binary quantity of whether the current estimate of the quantile is smaller or larger than the data point. This signal is used to estimate the true quantile using SGD.

The paper analyzes this algorithm theoretically to show that this allows them to derive an almost surely Gaussian approximation for the Polyak-Ruppert-type estimator of a quantile obtained by P-SGD and an almost surely consistent estimator of the quantile variance that relies only on the iterates of P-SGD.

Finally, the work also simulates some finite sample performance of the method using a Gaussian and Cauchy distributions which shows that they are able to achieve narrower intervals compared to previous works.

**Questions:**

Can the authors expand on how sensitive finite sample performance is on the choice of hyperparameters?

**Ethical Concerns:**

["NO or VERY MINOR ethics concerns only"]

**Final Justification:**

Based on the other reviews and rebuttals, I will choose to keep my initial rating.

**Limitations:**

yes

**Quality:**

3

**Strengths And Weaknesses:**

Strengths:
- Paper is easy to read and follow.
- The algorithm is simple, easy to implement, is fully online and requires low memory overhead.
- Work provides strong theoretical and simulation results.

Weaknesses:
- The algorithm might require tuning many hyperparameters

---

> ### Author Rebuttal · Authors · 2025-07-30
>
> We sincerely appreciate the time and effort you've invested in reviewing our paper. Your feedback is invaluable to us, and we have addressed each of your concerns below. Should you have any further questions, please do not hesitate to ask.
>
>
> **For weakness:**
>
> W1: Our tuning parameters fall into three categories. The first includes SGD-based parameters (e.g., the learning rate–related parameter $a$). Selecting the learning rate is indeed a well-known challenge in practice: SGD can be sensitive to this choice, especially in high-dimensional sparse settings and in rare–frequent or heavy-tailed regimes; see [1,2,3].
>
>
> We will reiterate this limitation in the revised manuscript and plan to investigate additional techniques for reducing our method's sensitivity to SGD-based parameters in future work. Nevertheless, under appropriate conditions, for example, when the objective is convex and smooth, or when the initialization is sufficiently close to the true parameter, Polyak–Ruppert averaged SGD enjoys provable convergence with tolerable sensitivity to the learning rate [4,5,6]. As later reported in our sensitivity studies, our results are robust to reasonable variations in this hyperparameter.
>
> The second category includes tuning parameters related to time-uniform inference, such as the AsympCS starting index $m$ and the hyperparameter $\rho$ in the Gaussian mixture bound (equation (6)). The third category consists of tuning parameters specific to our proposed method, such as the number of chains $h(t)$. We find that the results are not sensitive to these parameters; thus, recommendations from the time-uniform inference literature [7]  and the default setting provided in our paper (e.g., $h(t) = \lfloor 8 \log_{10}(t) \rfloor$) can serve as practical choices.
>
> Furthermore, according to your valuable comment, we have now conducted comprehensive sensitivity analyses for the aforementioned tuning parameters (i.e., $a$, $m$, $\rho$, and $h(t)$). Specifically, we consider one of the simulation settings from Section 4, with a total sample size of $T = 5{,}000{,}000$, $1,000$ repetitions, truthful response rate $r = 0.75$, quantile level $\tau = 0.5$, and normally distributed data. We evaluate the time-uniform type I error for AsympCS across a range of hyperparameter choices. The results are summarized in the following Tables. We are pleased to report that, the proposed methods maintain the nominal type I error rate (5\%) for nearly all hyperparameter choices, demonstrating its insensitivity to these tuning parameters.
>
> **Sensitivity analysis for $a$**: (t $\times 10^4$)
> | $a$ | method | t = 5 | t = 100 | t = 200 | t = 300 | t = 400 | t = 500 |
> |---|---------------|-----------|---------------|---------------|---------------|---------------|---------------|
> | 0.55 | UB | 0.003 | 0.005 | 0.005 | 0.005 | 0.005 | 0.005 |
> | 0.6 | UB | 0.002 | 0.003 | 0.003 | 0.003 | 0.003 | 0.004 |
> | 0.65 | UB | 0.002 | 0.003 | 0.004 | 0.004 | 0.004 | 0.004 |
> | 0.7 | UB | 0.002 | 0.002 | 0.002 | 0.002 | 0.002 | 0.002 |
> | 0.55 | GM | 0 | 0.002 | 0.005 | 0.01 | 0.012 | 0.014 |
> | 0.6 | GM | 0 | 0.001 | 0.002 | 0.004 | 0.009 | 0.012 |
> | 0.65 | GM | 0 | 0.002 | 0.007 | 0.01 | 0.012 | 0.014 |
> | 0.7 | GM | 0 | 0.001 | 0.002 | 0.003 | 0.007 | 0.009 |
>
>
> **Sensitivity analysis for $h(t)$** (t $\times 10^4$)
> | $h(t)$ | method | t = 5 | t = 100 | t = 200 | t = 300 | t = 400 | t = 500 |
> |---|---------------|-----------|---------------|---------------|---------------|---------------|---------------|
> | $6\log_{10}(t/5)$ | UB | 0.01| 0.016 | 0.017 | 0.018 | 0.019 | 0.02 |
> | $7\log_{10}(t/5)$ | UB | 0.006 | 0.008 | 0.01 | 0.011 | 0.011 | 0.011 |
> | $8\log_{10}(t/5)$ | UB | 0.002 | 0.003 | 0.003 | 0.003 | 0.003 | 0.004 |
> | $9\log_{10}(t/5)$ | UB | 0.003 | 0.003 | 0.003 | 0.003 | 0.003 | 0.003 |
> | $10\log_{10}(t/5)$ | UB | 0 | 0 | 0 | 0 | 0 | 0 |
> | $6\log_{10}(t/5)$ | GM | 0 | 0.006 | 0.025 | 0.043 | 0.054 | 0.065 |
> | $7\log_{10}(t/5)$ | GM | 0 | 0.001 | 0.012 | 0.021 | 0.027 | 0.031 |
> | $8\log_{10}(t/5)$ | GM | 0 | 0.001 | 0.002 | 0.004 | 0.009 | 0.012 |
> | $9\log_{10}(t/5)$ | GM | 0 | 0.001 | 0.004 | 0.006 | 0.007 | 0.01 |
> | $10\log_{10}(t/5)$ | GM | 0 | 0 | 0.001 | 0.002 | 0.004 | 0.004 |
>
>
> **Sensitivity analysis for $m$** (t $\times 10^4$)
> | $m$ | method |t = 100 | t = 200 | t = 300 | t = 400 | t = 500 |
> |---|---------------|---------------|---------------|---------------|---------------|---------------|
> | 1 | UB | 0.003 | 0.003 | 0.003 | 0.003 | 0.0035 |
> | 10000 | UB | 0.0065 | 0.007 | 0.0075 | 0.0075 | 0.008 |
> | 100000 | UB | 0.0065 | 0.008 | 0.008 | 0.008 | 0.0085 |
> | 500000 | UB | 0.0025 | 0.004 | 0.005 | 0.005 | 0.005 |
>
>
> **Sensitivity analysis for $\rho$** (t $\times 10^4$)
> | $\rho$ | method | t = 100 | t = 200 | t = 300 | t = 400 | t = 500 |
> |---|---------------|---------------|---------------|---------------|---------------|---------------|
> | 0.0005 | GM | 0 | 0 | 0 | 0 | 0.0005 | 0.0005 |
> | 0.001 | GM | 0.002 | 0.003 | 0.003 | 0.003 | 0.003 | 0.0035 |
> | 0.002 | GM | 0 | 0.0085 | 0.015 | 0.0185 | 0.0225 | 0.024 |
> | 0.005 | GM | 0 | 0.022 | 0.026 | 0.0275 | 0.029 | 0.029 |
>
>
>
> **Reference**
>
> [1] Linda F Wightman. Lsac national longitudinal bar passage study. lsac research report series. 1998.
>
> [2] Drago Ple\v{c}ko, Nicolas Bennett, and Nicolai Meinshausen. fairadapt: Causal reasoning for fair data preprocessing. Journal of Statistical Software, 110:1–35, 2024.
>
> [3] Jennifer Gillenwater, Matthew Joseph, and Alex Kulesza. Differentially privatequantiles. In International Conference on Machine Learning, pages 3713–3722. PMLR, 2021.
>
> [4] Boris T Polyak and Anatoli B Juditsky. Acceleration of stochastic approximation by averaging. SIAM journal on control and optimization, 30(4):838–855, 1992.
>
> [5]  Ilya Sutskever, James Martens, George Dahl, and Geoffrey Hinton. On the importance of initialization and momentum in deep learning. In International conference on machine learning, pages 1139–1147. pmlr, 2013.
>
>
> [6] Aymeric Dieuleveut and Francis Bach. Nonparametric stochastic approximation with large step-sizes. 2016.
>
> [7] Chuhan Xie, Kaicheng Jin, Jiadong Liang, and Zhihua Zhang. Asymptotic time-uniform inference for parameters in averaged stochastic approximation. arXiv preprint arXiv:2410.15057, 2024.

---

> > ### Author Response · Authors · 2025-08-06
> >
> > Thank you for your valuable time and positive feedback on our paper. We have added new experimental results to address your concern. If you have any further questions, please do not hesitate to ask.

---

> > > ### Comment · Reviewer_Nwru · 2025-08-08
> > >
> > > Thanks for the additional experiments. That clarifies all the questions I had. Please include the additional experiments in the paper.

---

> > > > ### Author Response · Authors · 2025-08-09
> > > >
> > > > Thank you for your encouraging feedback. We are pleased that our additional experiments have resolved your concerns. We shall include these results into the revision.

---

### Official Review · Reviewer_wd2F · 2025-07-03

**Clarity:** 3
**Significance:** 2
**Originality:** 2
**Rating:** 4
**Confidence:** 3

**Summary:**

The paper presents an online algorithm for computing time-uniform, asymptotic confidence sequences for quantiles under LDP.  Specifically, the proposed solution combines dynamically chained parallel SGD (P-SGD) with a randomized response (RR) mechanism to simultaneously estimate the target quantile and its variance. A Gaussian approximation leads to asymptotically anytime-valid confidence sequences whose widths obey the law of the iterated logarithm (LIL), with optimal convergence rates. The method is also computationally efficient, requiring only O(κ) memory for κ chains.

**Questions:**

Q1: The paper claims to provide the "first result on sequential inference for quantiles within the LDP framework". However, prior work like Liu et al. (2023) also presents an online LDP quantile inference method. Could you clarify the distinction?

Q2: Could you provide practical guidance on selecting h(t)?

Q3: The experimental results show that your P-SGD-based method, in the non-private setting (r=1), produces tighter confidence sequences than the state-of-the-art non-asymptotic method from Howard and Ramdas (2022). Could you explain the reason?

Q4: In theorem 2, it seems that big-O_a,s.(1) should be small-o?

**Ethical Concerns:**

["NO or VERY MINOR ethics concerns only"]

**Final Justification:**

In the rebuttal, the authors added real-world experiments (addressing W1). They also provided quantitative evidence for their efficiency claims (W3), as well as theoretical justification for their asymptotic approach (W4/Q3). The new experiment shows robustness to assumption violations (W5).

These changes have improved the quality and impact of the paper. The work now appears significantly more solid, well-evaluated, and technically sound than it did in the initial submission.

The remaining issues are the weak response regarding the hyperparameter h(t) (W2/Q2) and the unaddressed technical error in Theorem 2 (Q4).

**Limitations:**

Yes

**Quality:**

3

**Strengths And Weaknesses:**

Strengths:
S1. The paper introduces a novel approach that enables the consistent estimation of the estimator's variance without incurring any additional privacy cost.

S2. The paper provides a strong, almost sure Gaussian approximation for the quantile estimator.

S3. This work presents the first method for constructing time-uniform, asymptotic confidence sequences for quantiles with LDP.

Weaknesses:

W1. All experiments are conducted on synthetic data (mostly Normal and Cauchy). There is no demonstration on real-world datasets or application scenarios.

W2: The algorithm's performance relies on several key hyperparameters, most notably the function h(t) that determines the number of parallel chains. The specific choice used in the experiments (e.g., h(t)=⌊8log₁₀(t)⌋) is presented without justification or a sensitivity analysis showing how performance changes with different functions.

W3: The paper makes some empirical claims based on weak evidence. For instance, the claim that its pointwise confidence interval is "more efficient" than a competing method is supported only by visual inspection of a single plot (Figure 2), which is anecdotal.

W4. The confidence sequences are only asymptotically valid. There are no finite-sample coverage guarantees, which limits their practical reliability in early rounds.

W5. The theoretical results require a continuous and bounded derivative of the underlying density near the quantile. This excludes irregular or heavy-tailed distributions, despite claims of robustness.

---

> ### Author Rebuttal · Authors · 2025-07-30
>
> We sincerely appreciate the time and effort you've invested in reviewing our paper. Your feedback is invaluable to us, and we have addressed each of your concerns below. Should you have any further questions, please do not hesitate to ask.
>
>
> **For weakness:**
>
> W1: We have now incorporated two representative real datasets widely used in privacy research:
>
> Law school dataset [1]. This dataset consists of 20,649 examples aiming to predict students' undergraduate GPA based on their personal information and academic abilities. Given that GPA reflects individual educational outcomes and is protected under strict data-use agreements [1], we treat it as sensitive educational information requiring privacy protection.
>
> Government salary dataset [2]. This dataset originates from the 2018 American Community Survey conducted by the U.S. Census Bureau. It includes over 200,000 observations, with annual salary (USD) as the response variable. Annual salary represents typical personal financial information [3]; therefore, we treat it as sensitive data warranting privacy protection.
>
> To facilitate analysis, we applied a logarithmic transformation for two datasets and then back-transformed the confidence sequence bounds after prediction. We apply our proposed method: AsympCS based on equation (4) (UB) and equation (6) (GM)—to conduct privacy-preserving inference on the median ($\tau = 0.5$), targeting GPA in the first dataset and annual salary in the second. Specifically, we construct AsmpCS for the respective quantiles under truthful response rates $r = 0.75$ and $0.9$, and set the hyperparameter $\rho =0.01$ in equation (6) while keeping all other tuning parameters consistent with those used in Section 4. The upper and lower bounds of the confidence sequences are presented in Table 1. Due to space limitations, we present only the government salary dataset results; the GPA results of law school dataset are qualitatively similar.
>
> **Table 1.** AsmpCS for the government salary dataset.
> | Method | t=2000 | t=10000 | t=20000 | t=50000 | t=200000 |
> | :-: |-|-|-|-|-|
>  ||||**r = 0.75**|||
> | GM-L | 37898 | 44404 | 45774 | 47381  | 48326 |
> | GM-U | 49301 | 49097 | 48732 | 48850  | 48982 |
> | UB-L | 39522 | 44067 | 45348 | 47096  | 48197 |
> | UB-U | 47286 | 49472 | 49189 | 49145  | 49113 |
>  |||| **r = 0.9** |||
> | GM-L | 41707 | 45584 | 46375 |47728|48429|
> | GM-U | 50777 | 48980 | 48751 |48727|48954|
> | UB-L | 43036 | 45336 | 46030 |47533|48325|
> | UB-U | 49215 | 49247 | 49115 |48927|49059|
>
> From the results in Table 1, we observe that the CSs produced by our two methods (UB and GM) under different response rates $r$ covering similar central values. In addition, in both datasets, the length of the constructed CS decreases as $r$ increases. As $t$ grows, the sequence based on UB becomes more conservative than that based on GM. These observations align with our simulation findings and further demonstrate the methods' adaptability.
>
> W2:  According to your valuable comment, we have conducted comprehensive sensitivity analyses for several key hyperparameters. Due to the space limits, please see the rebuttal for reviewer By34.
>
> W3: We conduct additional experiments to compare our proposed confidence interval in Corollary 1 with the SN-based method in [9]. While both methods achieve empirical coverage rates close to the nominal confidence level, the average length of our confidence interval is more narrow across various settings of $r$ and $\tau$, indicating higher efficiency of our approach. Due to the space limits, we only present the following table.
>
> **Table 2.** $T = 1 \times 10^6$.
> | method | $\tau$ | $r = 1$   | $r = 0.9$  | $r = 0.75$ | $r = 0.5$  | $r = 0.25$ |
> |-|-|-|-|-|-|-|
> || | |  length | ($\times 10^{-3}$) | | |
> | SN-based| 0.8|7.22|8.33|10.8| 16.9| 35.7|
> || 0.5| 6.41| 7.03 | 8.47| 12.6| 26.2|
> || 0.3| 6.73 | 7.52| 9.20| 14.2| 29.3|
> |Ours| 0.8| 5.61| 6.59 | 8.48 | 13.8| 30.3|
> || 0.5| 4.89 | 5.48| 6.61|10.0| 21.3|
> || 0.3| 5.17| 5.86| 7.26| 11.3| 24.4|
>
> W4: In the concluding remarks of the original paper, we stated that “Finally, although non-asymptotic error bounds for SGD estimators have been extensively studied (e.g., [4]), extending these results to obtain fully non-asymptotic confidence sequences for SGD iterates under LDP remains an attractive yet challenging avenue for future research.”
>
> We elaborate on this point to clarify our results. Under local differential privacy, we estimate quantiles and conduct sequential inference using permutation-based SGD iterates, which are temporally dependent rather than i.i.d. Existing non-asymptotic sequential inference methods (e.g., [5]) assume i.i.d. data, and extending them to dependent settings remains challenging.
>
> Although our AsmpCSs are theoretically asymptotic, extensive experiments show they control type I error well, even in early rounds.
>
> Consider a special case when $r=1$, corresponding to the non-DP setting.  Our proposed method produces tighter confidence sequences than [6]. We elaborate on this comparison below for greater clarity.  Theoretically,  according to [5], the stitching function $\mathcal S_\alpha$ for the sub-gamma process with scale parameter $c$ is defined by,
>
> $\mathcal S_{\alpha}(v):=\sqrt{k_1^2v\ell(v)+k_2^2c^2\ell^2(v)}+k_2c\ell(v).$
>
> The non-asymptotic confidence sequence (CS) developed in [6] is based on the observation that a sub-Bernoulli process with range parameters $(g,h) = (p, 1-p)$ or $(g,h) = (1-p, p)$ is also sub-gamma with scale $c = (1-2p)/3$ or $c = (2p-1)/3$, respectively, as detailed in the proof of Theorem 1 therein.  In contrast, our asymptotic CS is based on a sub-Gaussian process, which corresponds to the special case where $c=0$,    which yields a sharper bound for $\mathcal{S}_\alpha(v)$. Consequently, the resulting confidence sequences are tighter than [6], and they  also align with the form of the law of the iterated logarithm, as discussed in [5].
>
> Empirically, our CSs are tighter and better calibrated (type I error near $\alpha$), while the non-asymptotic CS in [5] is often overly conservative. Similar empirical advantages of asymptotic CSs are also noted in [7].
>
> W5: Assumption (A1) ensures asymptotic normality of the sample quantile under the non-DP setting, as shown in Corollary 2.3.3.A of [8]. It requires that the distribution function $F(\cdot)$ is differentiable at the true quantile $x^\ast$, with a strictly positive derivative, in the non-DP framework. If $f(\cdot)$ is a density of $F(\cdot)$, it is not necessary that $f(\cdot)=F'(\cdot)$. However, if $f(\cdot)$ is continuous at $x^\ast$, then $F'(x^\ast)=f(x^\ast)>0$.
>
> Assumption (A2) is technical but important for controlling a negligible term in the Gaussian approximation, where a second-order Taylor expansion is used (see line 731). It is mild and holds for many distributions, including heavy-tailed ones like the Cauchy.
>
> For irregular distributions, we consider the density
>
> $f(x)= (\beta_{10,100}(x) + \beta_{100,100}(x) + \beta_{100,10}(x))/ 3,$
>
> where $\beta_{\alpha,\beta}(x)$ denotes the density of a Beta distribution with parameters $\alpha$ and $\beta$. This mixture has a   spike at $\tau = 0.5$, implying a large derivative of the density at that point. We perform additional numerical simulations and show the result under $r=0.75$ and $\tau=0.5$ in Table 3, confirming that our method remains valid.
>
> **Table 3.** $t \times 10^4$
> |method|t= 5|t= 100|t= 200|t= 300|t= 400|t = 500|
> |-|-|-|-|-|-|-|
> |UB|0.002|0.004|0.007|0.024|0.028| 0.029|
> |GM|0|0|0.002|0.031|0.036|0.046|
>
> **For questions:**
>
> Q1: We study asymptotic confidence sequences for quantiles under the LDP framework, while [9] developed pointwise confidence intervals. Confidence sequences provide valid intervals over time for ongoing monitoring, unlike confidence intervals which apply to fixed sample sizes. We estimate the variance via permutation-based SGD without extra privacy cost, whereas [9] uses a self-normalized method that avoids variance estimation but yields wider intervals.
>
> Q2: According to the sensitivity analysis for the choice of $h(\cdot)$ we conduct,
> the default setting provided in our original paper (e.g., $h(t) = \lfloor 8 \log_{10}(t) \rfloor$) can serve as a practical choice.
>
> Q3: Please see W4.
>
> Q4: Thank you for your careful reading. We would like to clarify that in Theorem 2, we indeed use $o_{a.s.}(1)$, not  $O_{a.s.}(1)$, as correctly stated in the manuscript. Kindly see page 6 for reference.
>
> **Reference**
> [1] Linda F Wightman. Lsac national longitudinal bar passage study. lsac research report series. 1998.
>
> [2] Drago Ple\v{c}ko, Nicolas Bennett, and Nicolai Meinshausen. fairadapt: Causal reasoning for fair data preprocessing. Journal of Statistical Software, 110:1–35, 2024.
>
> [3] Jennifer Gillenwater, Matthew Joseph, and Alex Kulesza. Differentially private quantiles. In International Conference on Machine Learning, pages 3713–3722. PMLR, 2021.
>
> [4] Likai Chen, Georg Keilbar, and Wei Biao Wu. Recursive quantile estimation: Non-asymptotic confidence bounds. Journal of Machine Learning Research, 24(91):1–25, 2023.
>
> [5] Steven R Howard, Aaditya Ramdas, Jon McAuliffe, and Jasjeet Sekhon. Time-uniform, nonparametric, nonasymptotic confidence sequences. The Annals of Statistics, 49(2):1055–1080, 2021.
>
> [6] Steven R Howard and Aaditya Ramdas. Sequential estimation of quantiles with applications to a/b testing and best-arm identification. Bernoulli, 28(3):1704– 1728, 2022.
>
> [7] Ian Waudby-Smith, David Arbour, Ritwik Sinha, Edward H Kennedy, and Aaditya Ramdas. Time-uniform central limit theory and asymptotic confidence sequences. The Annals of Statistics, 52(6):2613–2640, 2024.
>
> [8] Robert J Serfling. Approximation theorems of mathematical statistics. JohnWiley \& Sons, 2009.
>
> [9]  Yi Liu, Qirui Hu, Lei Ding, and Linglong Kong. Online local differential private quantile inference via self-normalization. In International Conference on Machine Learning, pages 21698–21714. PMLR, 2023.

---

> > ### Author Response · Authors · 2025-08-06
> >
> > We would like to extend our heartfelt gratitude for your valuable time and insightful feedback on our paper. We are committed to incorporating your suggestions to refine and enhance the clarity of our work. Should you have any further questions, please do not hesitate to ask. Thank you once again for your invaluable contributions to the development of our paper.

---

> > ### Comment · Reviewer_wd2F · 2025-08-07
> > **About results on the Law School and Government Salary datasets**
> >
> > Thank you for your clear and detailed response.
> >
> > One more question: your tables show the confidence sequence bounds at specific time points. Could you confirm if the primary claim of time-uniform coverage was also validated in these experiments? That is, did the empirical time-uniform type-I error remain below the nominal 5% level across all time steps for these real-world datasets, similar to the analysis in Figure 3 for synthetic data?

---

> > > ### Author Response · Authors · 2025-08-07
> > >
> > > Thank you for your follow up questions. For the real data analysis, it is difficult to verify the results through repeated experiments, as the entire algorithm can only be applied once, whereas the estimation of Type I error requires multiple replications. More importantly, since the underlying truth for real data is unknown, our results are mainly intended to serve as guidance for decision-making. We will include the figures rather than tables in the revision to enhance the presentation.

---

### Official Review · Reviewer_By34 · 2025-07-03

**Clarity:** 4
**Significance:** 2
**Originality:** 3
**Rating:** 5
**Confidence:** 3

**Summary:**

The paper studies the construction of asymptotic confidence sequence for the quantile of a distribution on the line under local differential privacy; where an $\epsilon,\delta$-differential privacy constraint applies to each individual datum in the sequence of observations.

The method is based on parallel private SGD, where the privacy mechanism is essentially randomized response and where the estimation of the quantile and its variance is done using the SGD iterates, where the parallel chains need to be constructed in a clever way". The authors provide both theoretical guarantees for their method as well as simulations.

**Questions:**

* What if the user is interested in more than just one quantile, but say more general tail risk? Can the method be adapted to these types of questions?
* What can be said about the method in terms of the efficiency theory (Steinberger 2024 - Efficiency in local differential privacy)?

**Ethical Concerns:**

["NO or VERY MINOR ethics concerns only"]

**Limitations:**

The main limitation I see are the hyperparameters of the method and setting these under differential privacy constraints in practice.

**Quality:**

3

**Strengths And Weaknesses:**

Strengths:
* I found the paper to be well organized and easy to read.
* The theoretical contributions are interesting independently of the motivation and are formulated as such.
* The method is well well explained - both in terms of its how and why.
* The writing is clear and the authors are forward in terms of what the do and don't assume.

Weakness:
* The method seems to rely on tuning parameters which are difficult to set in practice. Furthermore, it is unclear to me whether these tuning parameters can be set in a way that does not lead to "privacy leakage" due to having to run the method for different hyperparameter values.

A minor comment: sometimes there is some unnecessary notation overflow: On page 3, $U$ is used twice for completely different purposes. Similarly, the choice of the $\delta_{T,i}$-notation is somewhat unfortunate given the differential privacy definition.

---

> ### Author Rebuttal · Authors · 2025-07-30
>
> We sincerely appreciate the time and effort you've invested in reviewing our paper. Your feedback is invaluable to us, and we have addressed each of your concerns below. Should you have any further questions, please do not hesitate to ask.
>
>
> **For weakness:**
>
> Our tuning parameters fall into three categories.
>
> The first includes SGD-based parameters (e.g., the learning rate-related parameter $a$). Selecting the learning rate is indeed a well-known challenge in practice: SGD can be sensitive to this choice, especially in high-dimensional sparse settings and in rare–frequent or heavy-tailed regimes; see [1,2,3].
> We will reiterate this limitation in the revised manuscript and plan to investigate additional techniques for reducing our method's sensitivity to SGD-based parameters in future work. Nevertheless, under appropriate conditions, for example, when the objective is convex and smooth, or when the initialization is sufficiently close to the true parameter, Polyak–Ruppert averaged SGD enjoys provable convergence with tolerable sensitivity to the learning rate [4,5,6]. As later reported in our sensitivity studies, our results are robust to reasonable variations in this hyperparameter.
>
> The second category includes tuning parameters related to time-uniform inference, such as the AsympCS starting index $m$ and the hyperparameter $\rho$ in the Gaussian mixture bound (equation (6)).
>
> The third category consists of tuning parameters specific to our proposed method, such as the number of chains $h(t)$. We find that the results are not sensitive to these parameters; thus, recommendations from the time-uniform inference literature [7]  and the default setting provided in our paper (e.g., $h(t) = \lfloor 8 \log_{10}(t) \rfloor$) can serve as practical choices.
>
> Therefore,  we use a single, fixed set of tuning parameters in all our experiments, thereby avoiding scenarios in which multiple hyperparameter evaluations might inadvertently compromise privacy.
>
> Furthermore, according to your valuable comment, we have now conducted comprehensive sensitivity analyses for the aforementioned tuning parameters (i.e., $a$, $m$, $\rho$, and $h(t)$). Specifically, we consider one of the simulation settings from Section 4, with a total sample size of $T = 5{,}000{,}000$, $1000$ repetitions, truthful response rate $r = 0.75$, quantile level $\tau = 0.5$, and normally distributed data. We evaluate the time-uniform type I error for AsympCS across a range of hyperparameter choices. The results are summarized in the following Tables. We are pleased to report that, the proposed methods maintain the nominal type I error rate (5\%) for nearly all hyperparameter choices, demonstrating its insensitivity to these tuning parameters.
>
> **Sensitivity analysis for $a$**: (t $\times 10^4$)
> | $a$ | method | t = 5 | t = 100 | t = 200 | t = 300 | t = 400 | t = 500 |
> |---|---------------|-----------|---------------|---------------|---------------|---------------|---------------|
> | 0.55 | UB | 0.003 | 0.005 | 0.005 | 0.005 | 0.005 | 0.005 |
> | 0.6 | UB | 0.002 | 0.003 | 0.003 | 0.003 | 0.003 | 0.004 |
> | 0.65 | UB | 0.002 | 0.003 | 0.004 | 0.004 | 0.004 | 0.004 |
> | 0.7 | UB | 0.002 | 0.002 | 0.002 | 0.002 | 0.002 | 0.002 |
> | 0.55 | GM | 0 | 0.002 | 0.005 | 0.01 | 0.012 | 0.014 |
> | 0.6 | GM | 0 | 0.001 | 0.002 | 0.004 | 0.009 | 0.012 |
> | 0.65 | GM | 0 | 0.002 | 0.007 | 0.01 | 0.012 | 0.014 |
> | 0.7 | GM | 0 | 0.001 | 0.002 | 0.003 | 0.007 | 0.009 |
>
>
> **Sensitivity analysis for $h(t)$** (t $\times 10^4$)
> | $h(t)$ | method | t = 5 | t = 100 | t = 200 | t = 300 | t = 400 | t = 500 |
> |---|---------------|-----------|---------------|---------------|---------------|---------------|---------------|
> | $6\log_{10}(t/5)$ | UB | 0.01| 0.016 | 0.017 | 0.018 | 0.019 | 0.02 |
> | $7\log_{10}(t/5)$ | UB | 0.006 | 0.008 | 0.01 | 0.011 | 0.011 | 0.011 |
> | $8\log_{10}(t/5)$ | UB | 0.002 | 0.003 | 0.003 | 0.003 | 0.003 | 0.004 |
> | $9\log_{10}(t/5)$ | UB | 0.003 | 0.003 | 0.003 | 0.003 | 0.003 | 0.003 |
> | $10\log_{10}(t/5)$ | UB | 0 | 0 | 0 | 0 | 0 | 0 |
> | $6\log_{10}(t/5)$ | GM | 0 | 0.006 | 0.025 | 0.043 | 0.054 | 0.065 |
> | $7\log_{10}(t/5)$ | GM | 0 | 0.001 | 0.012 | 0.021 | 0.027 | 0.031 |
> | $8\log_{10}(t/5)$ | GM | 0 | 0.001 | 0.002 | 0.004 | 0.009 | 0.012 |
> | $9\log_{10}(t/5)$ | GM | 0 | 0.001 | 0.004 | 0.006 | 0.007 | 0.01 |
> | $10\log_{10}(t/5)$ | GM | 0 | 0 | 0.001 | 0.002 | 0.004 | 0.004 |
>
>
> **Sensitivity analysis for $m$** (t $\times 10^4$)
> | $m$ | method |t = 100 | t = 200 | t = 300 | t = 400 | t = 500 |
> |---|---------------|---------------|---------------|---------------|---------------|---------------|
> | 1 | UB | 0.003 | 0.003 | 0.003 | 0.003 | 0.0035 |
> | 10000 | UB | 0.0065 | 0.007 | 0.0075 | 0.0075 | 0.008 |
> | 100000 | UB | 0.0065 | 0.008 | 0.008 | 0.008 | 0.0085 |
> | 500000 | UB | 0.0025 | 0.004 | 0.005 | 0.005 | 0.005 |
>
>
>
> **Sensitivity analysis for $\rho$** (t $\times 10^4$)
> | $\rho$ | method | t = 100 | t = 200 | t = 300 | t = 400 | t = 500 |
> |---|---------------|---------------|---------------|---------------|---------------|---------------|
> | 0.0005 | GM | 0 | 0 | 0 | 0 | 0.0005 | 0.0005 |
> | 0.001 | GM | 0.002 | 0.003 | 0.003 | 0.003 | 0.003 | 0.0035 |
> | 0.002 | GM | 0 | 0.0085 | 0.015 | 0.0185 | 0.0225 | 0.024 |
> | 0.005 | GM | 0 | 0.022 | 0.026 | 0.0275 | 0.029 | 0.029 |
>
>
>
> **For minor comment:**
>
> To avoid a conflict with the symbol $U$ used in the upper bound in the confidence sequence, we have changed the notation for the random variable perturbing the gradient from $ U $ to $ W$ . Similarly, to prevent confusion with the symbol $\delta $ commonly used in the definition of differential privacy, we have replaced the radius of the confidence sequence with  $\gamma $.
>
>
>
> **For questions:**
>
>
> Q1: If we understand your question correctly, when estimating more than one quantile, such as multiple quantile values, a naive approach would be to divide the privacy budget and query the data multiple times. However, this method is inefficient and suffers from the quantile crossing issue. In the non-DP setting, there have been recent developments to address the crossing problem (see [8]). However, smoothing the gradient in these methods breaks the binary structure, potentially increasing the variance of the added noise under differential privacy.
>
> More generally, if the goal is to estimate functionals such as the distribution function (CDF) $P(X < t)$, the tail probability $P(X > t)$, or the conditional value at risk (CVaR) $\mathbb{E}(X \mathbf{1}(X > Q_{1-\alpha}) \mid X)$, then SGD-based approaches may not be applicable. For non-DP sequential inference of the CDF, see [9], and for local differential privacy (LDP) estimation of the CDF, see [10]. Sequential inference under LDP for these related functionals would require entirely different derivations, which we believe is an interesting but challenging direction.
>
>
> Q2: Since our main focus is sequential inference for quantiles, we only establish that the order of the proposed confidence sequence is optimal in the sense that it matches the law of the iterated logarithm (LIL) rate. However, we do not discuss the optimality of the average length of the sequence. The paper [11] you referred to investigates optimality for mean estimation and related problems under local differential privacy (LDP), as does the subsequent work [12].
>
> Regarding quantile estimation, SGD-based methods combined with binary permutation are shown to be optimal in some sense, as discussed in Section 3.5 of [13]. However, from an inference perspective, particularly in the sequential setting—obtaining an optimal confidence sequence remains challenging, even in the non-DP case.
>
>
>
> **Reference**
>
>
> [1] Léon Bottou, Frank E Curtis, and Jorge Nocedal. Optimization methods for
> large-scale machine learning. SIAM review, 60(2):223–311, 2018.
>
> [2] Jeremy Cohen, Simran Kaur, Yuanzhi Li, J Zico Kolter, and Ameet Talwalkar.
> Gradient descent on neural networks typically occurs at the edge of stability. In
> International Conference on Learning Representations, 2021.
>
> [3] John Duchi, Elad Hazan, and Yoram Singer. Adaptive subgradient methods
> for online learning and stochastic optimization. Journal of machine learning
> research, 12(7), 2011.
>
> [4] Boris T Polyak and Anatoli B Juditsky. Acceleration of stochastic approximation
> by averaging. SIAM journal on control and optimization, 30(4):838–855, 1992.
>
> [5]  Ilya Sutskever, James Martens, George Dahl, and Geoffrey Hinton. On the
> importance of initialization and momentum in deep learning. In International
> conference on machine learning, pages 1139–1147. pmlr, 2013.
>
>
> [6] Aymeric Dieuleveut and Francis Bach. Nonparametric stochastic approximation
> with large step-sizes. 2016.
>
> [7] Chuhan Xie, Kaicheng Jin, Jiadong Liang, and Zhihua Zhang. Asymptotic time-uniform inference for parameters in averaged stochastic approximation. arXiv
> preprint arXiv:2410.15057, 2024.
>
> [8] Chen, L., Keilbar, G., \& Wu, W. B. (2025). Smoothed SGD for quantiles: Bahadur representation and Gaussian approximation. arXiv preprint arXiv:2505.13299.
>
> [9]  Steven R Howard and Aaditya Ramdas. (2022) Sequential estimation of quantiles with
> applications to a/b testing and best-arm identification. Bernoulli, 28(3):1704–
> 1728.
>
> [10] Liu, Y., Hu, Q., \& Kong, L. (2024). Tuning-free estimation and inference of cumulative distribution function under local differential privacy.
>
>
> [11] Steinberger, L. (2024). Efficiency in local differential privacy. The Annals of Statistics, 52(5), 2139-2166.
>
>
> [12] Nikita, K., \& Steinberger, L. (2025, April). Efficient Estimation of a Gaussian Mean with Local Differential Privacy. In International Conference on Artificial Intelligence and Statistics (pp. 118-126). PMLR.
>
>
> [13] Liu, Y., Hu, Q., Ding, L., \& Kong, L. (2023, July). Online local differential private quantile inference via self-normalization. In International Conference on Machine Learning (pp. 21698-21714). PMLR.

---

> > ### Author Response · Authors · 2025-08-06
> >
> > Thank you for your expert analysis and constructive critique, which enhance the quality of our work. We have addressed each of your concerns. Please do not hesitate to ask if you have any further questions.

---

### Note · Authors · 2025-08-12

We thank the reviewers for their time and insightful feedback on our paper, which, to our knowledge, is the first to construct an asymptotic confidence sequence for quantile under LDP. While we appreciate the reviewers’ recognition of the work’s novelty, clarity, solid theoretical results, and practical algorithmic implementation, the reviewers also raised concerns regarding hyperparameter sensitivity, real-data applications, and broader aspects of our methodology and theory. We have addressed these points in the rebuttal and will incorporate the corresponding revisions into the manuscript, including a systematic sensitivity analysis, a real-data application, and expanded discussions of related literature, theory, and algorithms. We hope you find the revision satisfactory. Thank you again for your valuable feedback.

---

### Decision · Program_Chairs · 2025-09-17

**Decision:**

Accept (poster)

**Comment:**

## Metareview

This paper studies the question of private quantile inference for sequence of elements. Here we are given datapoints drawn from some underlying distribution, one at a time, and we would like to produce asymptotic confidence sequence (AsymCS) such that the true quantile lies in the sequence *simultaneously for all time step* with a certain probability. We would like to protect the datapoint via local differential privacy (LDP), meaning that each datapoint is privatized immediately without using central curator.

The main result of this work is an LDP algorithm which produces AsymCS such that the width of the confidence intervals is smaller than Law of Iterated Logarithm (LIL). This is obtained using a parallel SGD (P-SGD) approach where we construct multiple chains of (roughly) equal size, and each datapoint is allocated to a chain as they arrive. The actual SGD in each chain is done via a similar approach to [Liu et al., ICML 2023] who considers the pointwise version of the problem and uses binary randomized response to privatize each datapoint. Finally, the estimator is the weighted average of the chain-wise means.

## Strengths

- The setting of private AsymCS is natural in many modern scenarios e.g. in A/B testing. This paper provides a rigorous result that can be applied in such settings.

- This is the first work to use P-SGD in the context of private (asymptotic) confidence sequence. This is innovative and should be applicable to future work.

- The algorithm is simple, efficient (both from time and memory perspectives) and, thus, should be practical.

## Weaknesses

- Although this is the first work that considers computation of asymptotic confidence *sequence* with LDP, the *pointwise* setting has been considered before [Liu et al., ICML 2023] (who also used SGD-based algorithm with binary randomized response). However, getting anytime-valid guarantees for sequence like in this paper still requires significant amount of technical work.

- There are multiple hyperparameters that can be tuned and might affect the performance. This is addressed in the authors' response as they show sensitivity analysis that their AsymCS remains effective for a vast region of hyperparameters.

## Recommendation

Given that this is the first work to tackle private confidence sequences for quantiles and that it obtains very interesting rigorous results, we support acceptance.